# Live-cell single-molecule tracking reveals co-recognition of H3K27me3 and DNA targets polycomb Cbx7-PRC1 to chromatin

Chao Yu Zhen[1†], Roubina Tatavosian[1†], Thao Ngoc Huynh[1†], Huy Nguyen Duc[1†], Raibatak Das[2], Marko Kokotovic[1], Jonathan B Grimm[3], Luke D Lavis[3], Jun Lee[1], Frances J Mejia[1], Yang Li[1], Tingting Yao[4], Xiaojun Ren[1*]

[1]Department of Chemistry, University of Colorado Denver, Denver, United States; [2]Department of Integrative Biology, University of Colorado Denver, Denver, United States; [3]Janelia Research Campus, Howard Hughes Medical Institute, Ashburn, United States; [4]Department of Biochemistry and Molecular Biology, Colorado State University, Fort Collins, United States

**Abstract** The Polycomb PRC1 plays essential roles in development and disease pathogenesis. Targeting of PRC1 to chromatin is thought to be mediated by the Cbx family proteins (Cbx2/4/6/7/8) binding to histone H3 with a K27me3 modification (H3K27me3). Despite this prevailing view, the molecular mechanisms of targeting remain poorly understood. Here, by combining live-cell single-molecule tracking (SMT) and genetic engineering, we reveal that H3K27me3 contributes significantly to the targeting of Cbx7 and Cbx8 to chromatin, but less to Cbx2, Cbx4, and Cbx6. Genetic disruption of the complex formation of PRC1 facilitates the targeting of Cbx7 to chromatin. Biochemical analyses uncover that the CD and AT-hook-like (ATL) motif of Cbx7 constitute a functional DNA-binding unit. Live-cell SMT of Cbx7 mutants demonstrates that Cbx7 is targeted to chromatin by co-recognizing of H3K27me3 and DNA. Our data suggest a novel hierarchical cooperation mechanism by which histone modifications and DNA coordinate to target chromatin regulatory complexes.

*For correspondence: xiaojun.ren@ucdenver.edu

†These authors contributed equally to this work

## Introduction

Chemical covalent modification of histones and DNA regulates the chromatin structure states that play central roles in chromatin-templated biological processes (*Jenuwein and Allis, 2001*; *Li et al., 2007a*; *Luco et al., 2011*; *Ruthenburg et al., 2007*). This is exemplified by Polycomb group (PcG) proteins that function as histone-modifying enzymes and regulate gene expression *via* modulating higher order chromatin structures (*Simon and Kingston, 2013*). PcG proteins were initially identified as a body structure specification in *Drosophila* (*Lewis, 1978*). In mammals, PcG orthologs are essential for normal embryonic development and disease pathogenesis (*Helin and Dhanak, 2013*). For example, PcG subunits are frequently overexpressed or mutated in cancer, and perturbing PcG interactions can suppress cancer growth (*Helin and Dhanak, 2013*). Because of their clinical significance, enormous efforts have been devoted to develop drugs for targeting PcG subunits (*Helin and Dhanak, 2013*). However, the molecular mechanisms by which PcG proteins establish and maintain repressive Polycomb domains are still incompletely understood.

PcG proteins are generally found in one of two major protein complexes, the Polycomb repressive complex 1 or 2 (PRC1 or PRC2) (*Simon and Kingston, 2013*). PRC2 is a methyltransferase that

catalyzes di- and tri-methylation of lysine 27 on histone H3 (H3K27me2/3) by the SET domain of Ezh2 (or Ezh1) (*Cao et al., 2002*; *Czermin et al., 2002*; *Kuzmichev et al., 2002*; *Margueron et al., 2008*; *Muller et al., 2002*; *Shen et al., 2008*). Unlike most SET domain methyltransferases, Ezh2 requires Suz12 and Eed for enzymatic activity (*Cao and Zhang, 2004*; *Martin et al., 2006*; *Montgomery et al., 2005*; *Pasini et al., 2004*). Additionally, Rbbp4 and Rbbp7 are stoichiometric subunits of PRC2 (*Cao et al., 2002*; *Cao and Zhang, 2004*; *Margueron and Reinberg, 2011*). In contrast, PRC1 is an ubiquitin ligase that monoubiquitylates histone H2A on lysine 119 (H2AK119ub1) (*de Napoles et al., 2004*; *Wang et al., 2004a*). PRC1 complexes form around Ring1b (or Ring1a) subunits with which one of the six Pcgf proteins (Pcgf1-6) associates (*Gao et al., 2012*; *Gil and O'Loghlen, 2014*; *Tavares et al., 2012*). The Ring-Pcgf2 (Mel18) or Pcgf4 (Bmi1) hetero-dimers are incorporated in canonical PRC1 (Cbx-PRC1; the functional homolog to *Drosophila* PRC1) and the other Ring-Pcgf heterodimers are assembled in variant PRC1 (vPRC1). The Cbx-PRC1 complex is composed of one of each of four different core subunits, Ring1 (Ring1a/Ring1b), Pcgf (Mel18/Bmi1), Phc (Phc1/2/3), and Cbx (Cbx2/4/6/7/8). In contrast, the vPRC1 complexes contain Rybp or Yaf instead of Cbx and Phc.

Several mechanisms underlying the targeting of PRC1 to chromatin have been documented (*Blackledge et al., 2015*; *Simon and Kingston, 2013*). Initial studies of *Drosophila* PcG (dPcG) proteins have suggested a mechanism of the PRC2-mediated recruitment of PRC1 (*Cao et al., 2002*; *Min et al., 2003*; *Wang et al., 2004b*). dPRC2 is recruited to Polycomb response elements (PRE) by its interaction with sequence-specific DNA-binding proteins and then modifies chromatin with H3K27me3 that recruits dPRC1. Consistent with the notion, genetic analyses have demonstrated that dPRC1 and dPRC2 co-regulate PcG target genes and dPRC1 is displaced from chromatin in dPRC2 mutants (*Cao et al., 2002*; *Wang et al., 2004b*). Genome-wide studies have shown that dPRC1 and dPRC2 co-occupy many PcG target genes (*Schwartz et al., 2006*).

In mammals, the recruitment of PRC1 is enigmatic and complicated, and has been broadly defined as H3K27me3-dependent and –independent recruitment mechanisms (*Blackledge et al., 2015*; *Farcas et al., 2012*; *He et al., 2013*; *Tavares et al., 2012*). An additional layer of complexity is added when considering that PRC1, in some cases, recruits PRC2 (*Blackledge et al., 2014*; *Cooper et al., 2014*; *Kalb et al., 2014*). The H3K27me3-dependent recruitment of mammalian PRC1 originates from the *Drosophila* model and is based on the facts that the Cbx family members and dPc both contain a conserved chromodomain (CD) (*Blackledge et al., 2015*). The model is consistent with studies demonstrating a link between H3K27me3 and PRC1 recruitment (*Agger et al., 2007*; *Boyer et al., 2006*; *Lee et al., 2007*; *Mujtaba et al., 2008*). Although the model for the mammalian Cbx-PRC1 recruitment is prevalent, several lines of evidence argue against the proposed model as a general mechanism of action. First, unlike to the dPc CD, the Cbx CDs have a much weaker affinity for H3K27me3 (*Bernstein et al., 2006*; *Kaustov et al., 2011*; *Tardat et al., 2015*). The Cbx2 CD shows preference for H3K27me3 while the Cbx4 and Cbx7 CDs exhibit preference for H3K9me3 (*Bernstein et al., 2006*; *Kaustov et al., 2011*; *Tardat et al., 2015*). The affinity of the Cbx6 and Cbx8 CDs for H3K27me3 is nearly undetectable (*Bernstein et al., 2006*; *Kaustov et al., 2011*). One question is whether the recognition of H3K27me3 by the Cbx CDs is required for the targeting of Cbx proteins to chromatin. Likewise, genome-wide approaches have demonstrated that H3K27me3 forms a broad domain and the binding PRC1 is sharply localized within the H3K27me3 domain, and that a subset of H3K27me3 domains corresponds to PRC1 binding sites (*Ku et al., 2008*). Thus, there are missing molecular links between genetic, biochemical, and genome-wide analysis for our understanding of how the Cbx-PRC1 complexes are targeted to chromatin.

Single-molecule techniques have been widely applied to study DNA- and chromatin-templated processes *in vitro* and provide insights into genetic information flow *in vivo* (*Bell and Kowalczykowski, 2016*; *Dangkulwanich et al., 2014*; *Duzdevich et al., 2014*; *Geertsema and van Oijen, 2013*; *Harada et al., 2016*; *Herbert et al., 2008*; *Li et al., 2004*; *Ngo et al., 2015*; *Ren et al., 2003*; *Ren et al., 2006*; *Tatavosian et al., 2015*). Recent advances in single-molecule imaging allow measuring the quantitative kinetics of gene control in living mammalian cells (*Chen et al., 2014*; *Coleman et al., 2015*; *Gebhardt et al., 2013*; *Grimm et al., 2015*; *Izeddin et al., 2014*; *Katz et al., 2016*; *Knight et al., 2015*; *Liu et al., 2015, 2014*; *Mazza et al., 2012, 2013*; *Morisaki et al., 2014*; *Normanno et al., 2015*; *Swinstead et al., 2016*; *Zhang et al., 2014*). Here, we combine live-cell SMT and genetic engineering to determine whether H3K27me3 is required for the targeting of Cbx proteins to chromatin and to dissect the targeting mechanisms. Single-molecule quantitative

measurement is used to determine the kinetics and dynamics of the Cbx protein interactions with chromatin in living mouse embryonic stem (mES) cells. The analyses demonstrate a new functional role of the Cbx-PRC1 complex formation in the targeting of Cbx7 to chromatin and uncover the molecular mechanism underlying the targeting of Cbx7 to chromatin and fill in the knowledge gap between genetic, biochemical, and genome-wide analyses. These results contribute significantly to our quantitative understanding of kinetics and dynamics of the Cbx-PRC1 proteins in living cells, allowing us to suggest the molecular mechanisms underlying how the Cbx-PRC1 complexes are targeted to chromatin.

## Results

### Validation of live-cell SMT using HaloTag and histone H2A fused to HaloTag

To investigate the Cbx proteins binding dynamics at endogenous genomic loci, we performed SMT to determine diffusion and chromatin binding properties of individually fluorescently labeled Cbx molecules within living mES cells. HaloTag was fused at the N-terminus of Cbx proteins under an inducible, tetracycline response element (TRE)-tight promoter (*Figure 1A*). These fusion genes were stably integrated into the genome of wild-type (PGK12.1) mES cells. We used highly inclined thin illumination (HILO) to avoid stray-light reflection and to reduce background from cell auto-fluorescence (*Tokunaga et al., 2008*) (*Figure 1B*). The HaloTag ligand of the bright, photostable fluorophore Janelia Fluor 549 (JF$_{549}$) allowed for visualization of single HaloTag-Cbx molecules at their basal expression level without doxycycline induction (*Grimm et al., 2015*) (*Figure 1C*).

To validate our live-cell SMT system, we investigated HaloTag-NLS (NLS, nucleus localization sequence) and H2A-HaloTag. A visual inspection of single-molecule imaging tracks showed Halo-Tag-NLS and H2A-HaloTag exhibited obvious differences (*Video 1* and *2*). The majority of H2A-HaloTag molecules were stationary while almost every HaloTag-NLS molecules were mobile. Analysis of the distributions of track length for HaloTag labelled molecules indicated that ~15–35% of the tracks have a frame number $\leq$ 3 (*Figure 1—figure supplement 1A*). To avoid bias toward slowly moving molecules, we calculated the maximum likelihood diffusion coefficient ($D_m$) per track during a fixed time interval of 30 ms and constructed the $logD_m$ distributions for HaloTag-NLS and H2A-HaloTag (*Figure 1D* and *Figure 1—source data 1*). The histograms for H2A-HaloTag were fitted with three populations (see Materials and methods): F$_1$ ($D_{m1}$), F$_2$ ($D_{m2}$), and F$_3$ ($D_{m3}$). We measured F$_1$ = (72 ± 1)% ($D_{m1}$ = 0.032 ± 0.001 μm$^2$s$^{-1}$), F$_2$ = (14 ± 2)% ($D_{m2}$ = 0.50 ± 0.06 μm$^2$s$^{-1}$), and F$_3$ = (14 ± 2)% ($D_{m3}$ = 2.4 ± 0.2 μm$^2$s$^{-1}$) (*Figure 1F* and *Supplementary file 1*). The histograms for HaloTag-NLS were fitted with two populations since the fitting with a three-component Gaussian function using the fixed value $D_{m1}$ = 0.032 μm$^2$s$^{-1}$ did not converge (see Materials and methods): F$_2$ ($D_{m2}$) and F$_3$ ($D_{m3}$). We measured F$_2$ = (71 ± 1)% ($D_{m2}$ = 0.24 ± 0.03 μm$^2$s$^{-1}$) and F$_3$ = (29 ± 2)% ($D_{m3}$ = 2.5 ± 0.1 μm$^2$s$^{-1}$) (*Figure 1F* and *Supplementary file 1*). We designated the F$_1$ component as the chromatin-bound (CB) population, whose slow motion reflects the chromosomal dynamics and measurement uncertainties, F$_2$ as the intermediate diffusion (ID) population, whose motion reflects non-specific interaction with chromatin or confined movement, and F$_3$ as the fast diffusion (FD) population, whose motion reflects freely diffusing molecules. Experimental evidence supports the assignment of the F$_1$ component as the CB population because (1) the CB population of HaloTag-NLS is almost undetectable, (2) ~70% of H2A-HaloTag molecules are incorporated into chromatin, which is consistent with the reported results from fluorescence photobleaching after recovery (*Ren et al., 2008*), and (3) the diffusion constant and fractional size of the CB population of H2A-HaloTag agree with recent single-molecule reports (*Gebhardt et al., 2013*; *Mazza et al., 2012*).

The above $D_m$ analysis involves averaging over independent pairs of the squared jump distance of a single trajectory with a 30-ms interval. Such averaging might obscure transitions between chromatin-binding, confined, and Brownian motion for the single trajectory of a particle within the observation time. To investigate whether the averaging affects resolving the kinetic fractions, we calculated $D_{f1}$ based on the squared jump distance between the initial position $r_0$ and the first position $D_m$ of a single-trajectory with a 30-ms interval, and constructed the $logD_{f1}$ distribution (*Figure 1—figure supplement 1B* and *Figure 1—source data 1*). Counting only the first displacement of each track has been reported previously for studying of transcription factors binding to DNA (*Gebhardt et al.,*

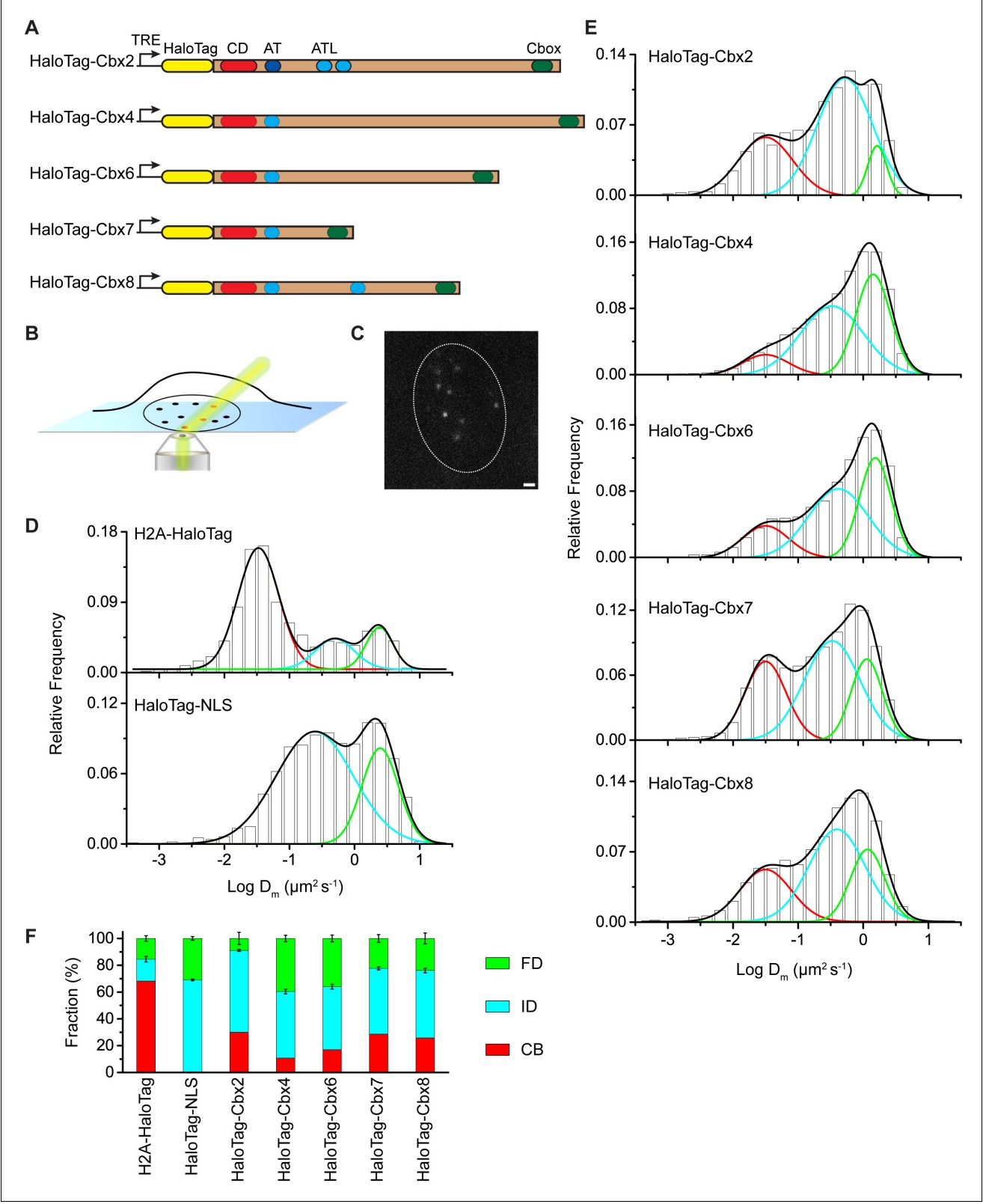

**Figure 1.** The Cbx family members exhibit distinct dynamics in living mES cells. (**A**) The sequences encoding the five Cbx proteins were fused with HaloTag to generate the HaloTag-Cbx fusions that were stably expressed in wild-type (PGK12.1) mES cells. The expression level of HaloTag-Cbx fusions was controlled by Tet-responsive element (TRE). HaloTag is shown in yellow, CD (chromodomain) in red, AT (AT-hook) motif in light blue; ATL (AT-hook-like) motif in cyan, and Cbox (chromobox) in emerald. (**B**) Schematic representation of highly inclined and laminated optical sheet (HILO)

*Figure 1 continued on next page*

*Figure 1 continued*

microscopy. (**C**) Live-cell single-molecule visualization of HaloTag-Cbx7 molecules in mES cells during a 30-ms exposure. Oval white dash circle outlines the nucleus of the cell. The individual white points represent single HaloTag-Cbx7 molecules. Scale bar, 2 μm. (**D**) Normalized histograms of the log maximum likelihood diffusion coefficient $D_m$ for H2A-HaloTag (N = 19 cells, n = 2675 trajectories) and HaloTag-NLS (N = 69 cells, n = 2087 trajectories) in wild-type mES cells. The H2A-HaloTag histogram was fitted with a three-component Gaussian and the HaloTag-NLS histogram a two-component Gaussian. The color bars indicate that the fraction of proteins in the chromatin-bound (CB, red), intermediate (ID, cyan), and fast diffusion (FD, green) population. NLS, nuclear localization sequence. (**E**) Normalized histograms of the log maximum likelihood diffusion coefficient $D_m$ for HaloTag-Cbx2 (N = 44 cells, n = 2833 trajectories), HaloTag-Cbx4 (N = 34 cells, n = 11,343 trajectories), HaloTag-Cbx6 (N = 33 cells, n = 7457 trajectories), HaloTag-Cbx7 (N = 51 cells, n = 3097 trajectories), and HaloTag-Cbx8 (N = 36 cells, n = 3351 trajectories) in wild-type mES cells. The histograms were fitted with a three-component Gaussian. (**F**) Fraction of the CB (red), ID (cyan), and FD (green) population for H2A-HaloTag, HaloTag-NLS, HaloTag-Cbx2, HaloTag-Cbx4, HaloTag-Cbx6, HaloTag-Cbx7, and HaloTag-Cbx8. The data were obtained from *Figure 1D and E* fitted with a Gaussian. Results are means ± SD.

The following source data and figure supplements are available for figure 1:

**Source data 1.** Source data for *Figure 1D–E* and *Figure 1—figure supplement 1B* and *2A*.

**Figure supplement 1.** Comparison of the $D_m$ and $D_{f1}$ analysis.

**Figure supplement 2.** Control experiments for testing the effects of the endogenous Cbx7 protein on the kinetic fractions of the exogenous HaloTag-Cbx7 fusion.

**Figure supplement 3.** Control experiments for analyzing the protein level of HaloTag-Cbx7 and for testing whether HaloTag-Cbx7 occupies Polycomb target promoters.

*2013*). The fractional sizes of the individual populations obtained from the $D_{f1}$ analysis were comparable to those obtained from the $D_m$ analysis (*Figure 1—figure supplement 1C*). Since a typical $D_m$ histogram visually resolved populations better than a typical $D_{f1}$ histogram (*Figure 1D–E* and *Figure 1—figure supplement 1B*), we performed our analysis using the $D_m$ analysis throughout the text.

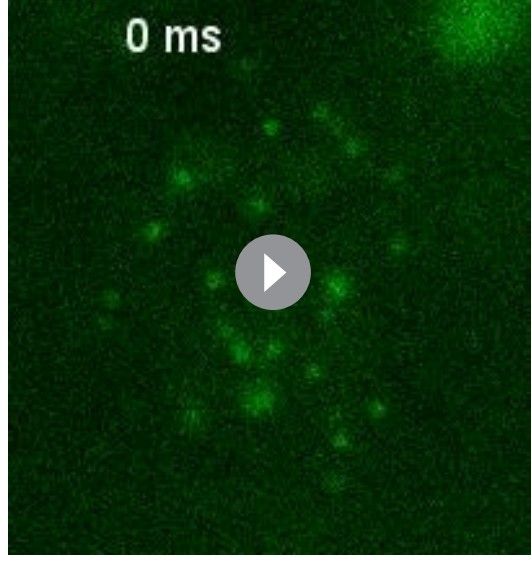

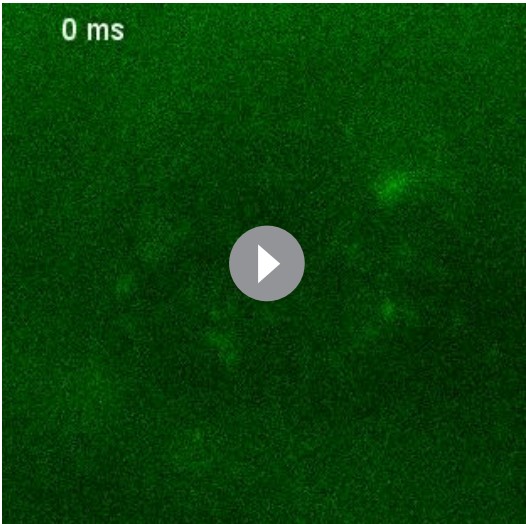

**Video 1.** H2A-HaloTag in wild-type mES cells (Fractional studies).

**Video 2.** HaloTag-NLS in wild-type mES cells (Fractional studies).

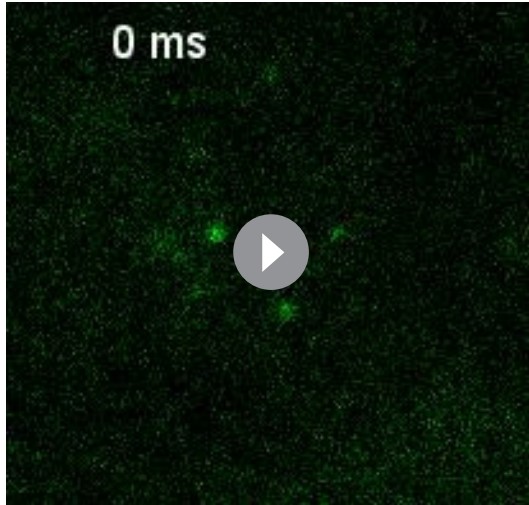

**Video 3.** HaloTag-Cbx2 in wild-type mES cells (Fractional studies).

**Video 4.** HaloTag-Cbx4 in wild-type mES cells (Fractional studies).

## Distinct chromatin-binding behaviors among the Cbx proteins

At the single-molecule level, we quantitatively measured diffusion constants and chromatin-binding levels of the Cbx proteins in mES cells (*Video 3—7*). We fitted the histograms with a three-component Gaussian function and calculated the diffusion constants and the fractional sizes of the individual populations (*Figure 1E* and *Figure 1—source data 1*). Since the peak centers of the CB populations for the HaloTag-Cbx proteins were almost the same as that for H2A-HaloTag, we fixed $logD_{m1}$ to be the value $-1.5$ ($D_{m1}$= 0.032 μm²s⁻¹)(see Materials and methods). We measured $F_1$ = (30 ± 1)%, $F_2$ = (61 ± 1)%, and $F_3$ = (9 ± 4)% for HaloTag-Cbx2, $F_1$ = (10 ± 1)%, $F_2$ = (50 ± 1)%, and $F_3$ = (40 ± 2)% for HaloTag-Cbx4, $F_1$ = (17 ± 1)%, $F_2$ = (47 ± 2)%, and $F_3$ = (36 ± 3)% for HaloTag-Cbx6, $F_1$ = (29 ± 1)%, $F_2$ = (49 ± 1)%, and $F_3$ = (22 ± 3)% for HaloTag-Cbx7, and $F_1$ = (26 ± 1)%, $F_2$ = (50 ± 2)%, and $F_3$ = (24 ± 4)% for HaloTag-Cbx8 (*Figure 1F* and *Supplementary file 1*). To complement

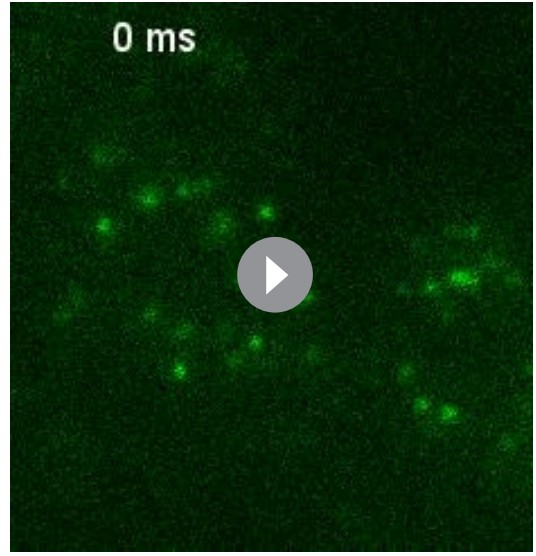

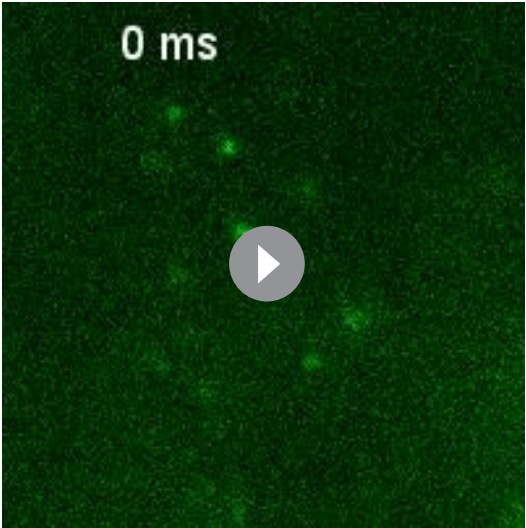

**Video 5.** HaloTag-Cbx6 in wild-type mES cells (Fractional studies).

**Video 6.** HaloTag-Cbx7 in wild-type mES cells (Fractional studies).

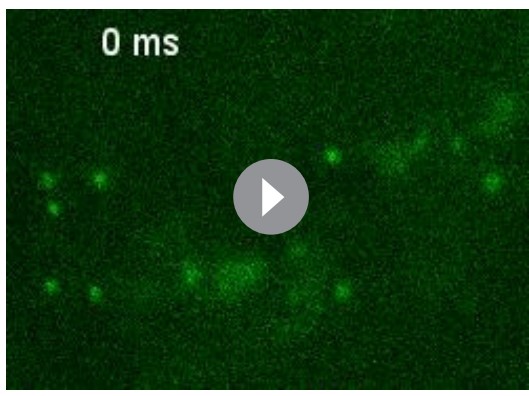

**Video 7.** HaloTag-Cbx8 in wild-type mES cells
(Fractional studies).

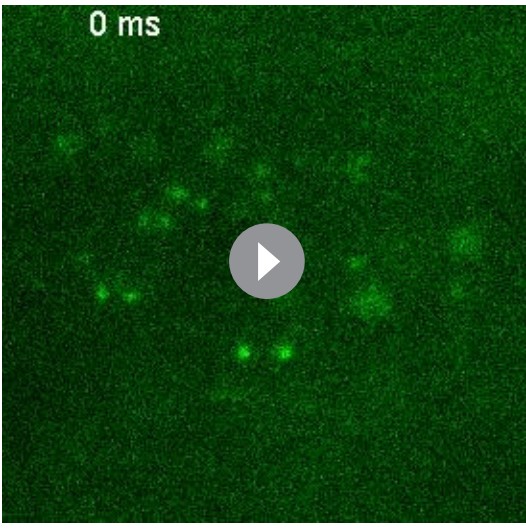

**Video 8.** HaloTag-Cbx7 in *Cbx7* KO mES cells
(Fractional studies).

the $D_m$ analysis, we also performed the $D_{f1}$ analysis for HaloTag-Cbx7 (*Figure 1—figure supplement 1B* and *Figure 1—source data 1*). The fractional sizes obtained from the $D_{f1}$ analysis were comparable to those obtained from the $D_m$ analysis (*Figure 1—figure supplement 1C*). These data provided a few novel observations: (1) the Cbx proteins exhibit distinct chromatin-associating capacities, (2) Cbx2, Cbx7, and Cbx8 exhibit the highest chromatin-bound level while Cbx4 has the lowest one, (3) the fractional sizes of the FD components are distinct among the Cbx family proteins, (4) except for Cbx2, the fractional sizes of the ID components are similar among the Cbx proteins, and (5) among the Cbx proteins, the diffusion constants are distinct for the ID components, but similar for the FD components. Altogether, our results demonstrate that the Cbx proteins employ distinct ways to interact with chromatin and to explore the nucleus.

The above SMT experiments were performed in wild-type mES cells where the endogenous and exogenous (fusion) proteins co-exist. HaloTag may make the fusion proteins less-equal competition with their endogenous counterparts. Given that Cbx7 is the major Cbx protein within mES cells (*Morey et al., 2013*; *Morey et al., 2012*), we integrated *HaloTag-Cbx7* to the genome of $Cbx7^{-/-}$ mES cells and performed SMT (*Video 8*). We measured $F_1 = (30 \pm 1)\%$, $F_2 = (46 \pm 1)\%$, and $F_3 = (24 \pm 2)\%$ for HaloTag-Cbx7, which are comparable to those obtained from HaloTag-Cbx7 in wild-type mES cells (*Figure 1—figure supplement 2A–B*, *Supplementary file 1*, and *Figure 1—source data 1*). Next, we performed biochemical analysis of Cbx7. Immunoblotting indicated that the level of HaloTag-Cbx7 protein was less than that of its endogenous counterpart (*Figure 1—figure supplement 3A*). Chromatin immunoprecipitation (ChIP) analysis indicated that HaloTag antibody greatly precipitated promoters of Polycomb target genes from *HaloTag-Cbx7/Cbx7$^{-/-}$* mES cells, but much less from wild-type mES cells (*Figure 1—figure supplement 3B*), suggesting that the HaloTag-Cbx7 protein binds to Polycomb target genes.

## H3K27me3 is important for the targeting of Cbx7 and Cbx8 to chromatin, but plays a less important role for Cbx2, Cbx4, and Cbx6

To investigate if H3K27me3 is required for the targeting of Cbx proteins to chromatin within mES cells (*Figure 2—figure supplement 1A*), we integrated *HaloTag-Cbx* fusion genes into the genome of $Eed^{-/-}$ mES cells. Eed is a core component of PRC2 (*Margueron and Reinberg, 2011*). H3K27me3 was almost undetectable in $Eed^{-/-}$ mES cells (*Figure 2E*). We performed SMT of HaloTag-Cbx proteins in $Eed^{-/-}$ mES cells (*Video 9—13*). The $logD_m$ histograms for HaloTag-Cbx2, HaloTag-Cbx4, HaloTag-Cbx6, and HaloTag-Cbx8 were fitted with three populations (*Figure 2A* and *Figure 2—source data 1*). The $logD_m$ histograms for HaloTag-Cbx7 were fitted with two populations rather than three populations since the fitting with a three-component Gaussian function using the fixed value $D_{m1} = 0.032 \ \mu m^2 s^{-1}$ did not converge (see Materials and methods). We measured $F_1 =$

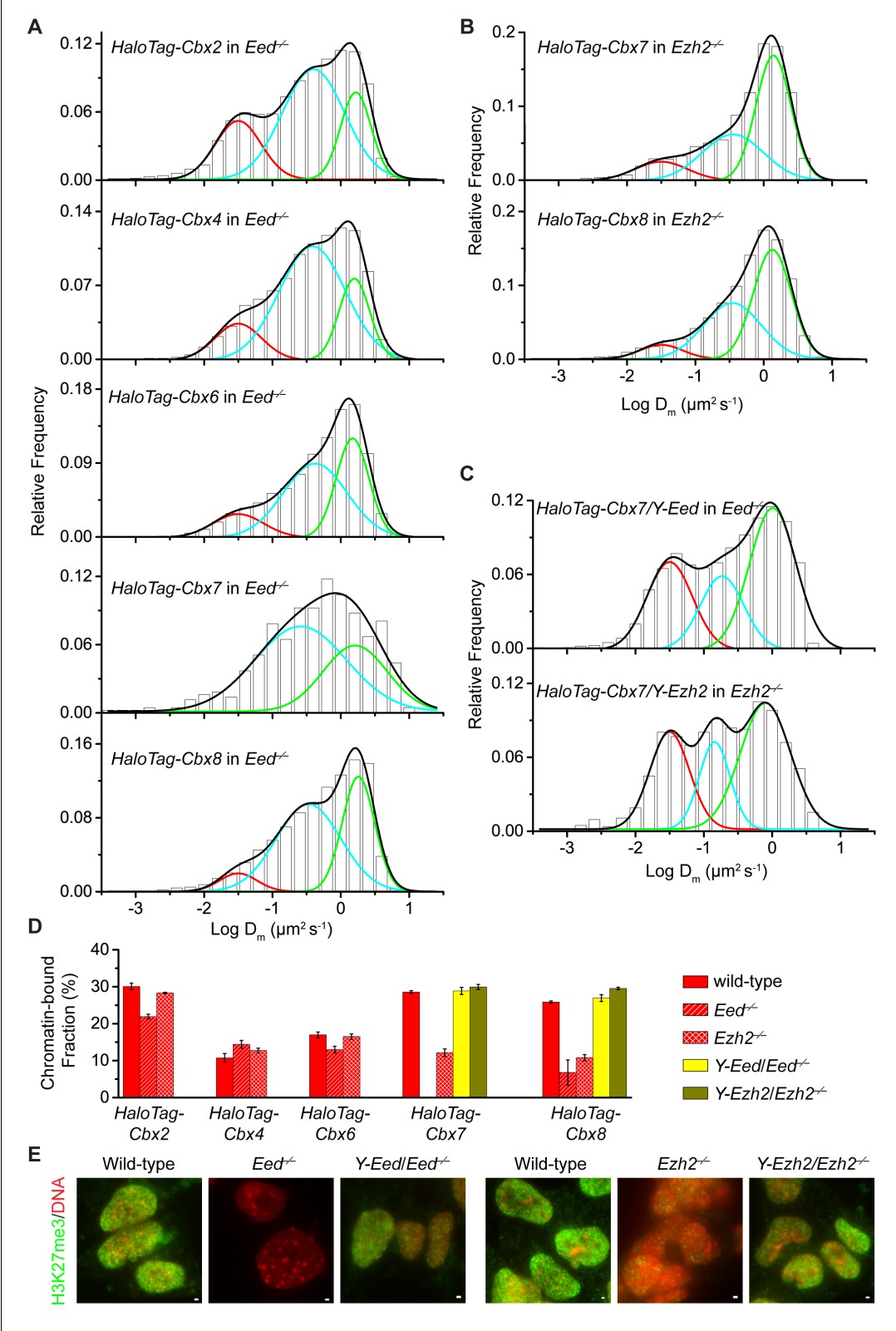

**Figure 2.** H3K27me3 is important for the targeting of Cbx7 and Cbx8 to chromatin, but plays a less important role for Cbx2, Cbx4, and Cbx6. (**A**) Normalized histograms of the log maximum likelihood diffusion coefficient $logD_m$ for HaloTag-Cbx2 (N = 27 cells, n = 2471 trajectories), HaloTag-Cbx4 (N = 21 cells, n = 3254 trajectories), HaloTag-Cbx6 (N = 11 cells, n = 4860 trajectories), HaloTag-Cbx7 (N = 25 cells, n = 453 trajectories), and HaloTag-Cbx8 (N = 47 cells, n = 5825 trajectories) in $Eed^{-/-}$ mES cells. The distributions for HaloTag-Cbx2, HaloTag-Cbx4, HaloTag-Cbx6, and HaloTag-Cbx8

*Figure 2 continued on next page*

*Figure 2 continued*

were fitted with three populations while the distribution for HaloTag-Cbx7 with two populations. (B) Normalized histograms of the log maximum likelihood diffusion coefficient $logD_m$ for HaloTag-Cbx7 (N = 26 cells, n = 3874 trajectories) and HaloTag-Cbx8 (N = 42 cells, n = 9220 trajectories) in $Ezh2^{-/-}$ mES cells. The distributions were fitted with three components. (C) Normalized histograms of the log maximum likelihood diffusion coefficient $logD_m$ for HaloTag-Cbx7 in $Y$-$Eed/Eed^{-/-}$ (N = 16 cells, n = 1733 trajectories) and $Y$-$Ezh2/Ezh2^{-/-}$ (N = 14 cells, n = 846 trajectories) mES cells. The histograms were fitted with a three-component Gaussian. (D) Chromatin-bound fraction for HaloTag-Cbx2, HaloTag-Cbx4, HaloTag-Cbx6, HaloTag-Cbx7, and HaloTag-Cbx8 in wild-type (red solid), $Eed^{-/-}$ (red strip), and $Ezh2^{-/-}$ (red cross-strip) mES cells, and for HaloTag-Cbx7 and HaloTag-Cbx8 in $Y$-$Eed/Eed^{-/-}$ (yellow solid) and $Y$-$Ezh2/Ezh2^{-/-}$ (dark yellow solid) mES cells. The data were obtained from *Figure 1E*, *Figure 2A–C*, and *Figure 2—figure supplement 1B–C* fitted with a Gaussian function. Results are means ± SD. (E) Immunostaining of H3K27me3 in wild-type, $Eed^{-/-}$, $Ezh2^{-/-}$, $Y$-$Eed/Eed^{-/-}$, and $Y$-$Ezh2/Ezh2^{-/-}$ mES cells by using antibody directed against H3K27me3 (green). DNA was stained with hoechst (red). Overlay images are shown. Note that H3K27me3 staining is visible in $Ezh2^{-/-}$ mES cells because of the redundancy of Ezh1. Scale bar is 5 μm.

The following source data and figure supplement are available for figure 2:

**Source data 1.** Source data for *Figure 2A–C* and *Figure 2—figure supplementary 1B–C*.

**Figure supplement 1.** Additional experiments for HaloTag-Cbx in $Eed^{-/-}$ and $Ezh2^{-/-}$ mES cells.

(26 ± 1)%, $F_2$ = (54 ± 1)%, and $F_3$ = (20 ± 2)% for HaloTag-Cbx2, $F_1$ = (14 ± 1)%, $F_2$ = (65 ± 1)%, and $F_3$ = (21 ± 2)% for HaloTag-Cbx4, $F_1$ = (13 ± 1)%, $F_2$ = (52 ± 1)%, and $F_3$ = (35 ± 2)% for HaloTag-Cbx6, $F_2$ = (65 ± 3)% and $F_3$ = (35 ± 6)% for HaloTag-Cbx7, and $F_1$ = (8 ± 2)%, $F_2$ = (40 ± 1)% and $F_3$ = (52 ± 1)% for HaloTag-Cbx8 (*Figure 2D* and *Supplementary file 1*). By comparing these results to those obtained from wild-type mES cells, two conclusions could be made: (1) the CB components of Cbx7 and Cbx8 are either nearly undetectable or significantly reduced, suggesting that H3K27me3 contributes significantly to the targeting of Cbx7 and Cbx8 to chromatin, and (2) the levels of the CB components for Cbx2, Cbx4, and Cbx6 are similar or slightly reduced, suggesting that in contrast to Cbx7 and Cbx8, H3K27me3 plays a less important role in the targeting of Cbx2, Cbx4, and Cbx6 to chromatin. Taken together, our results demonstrate that H3K27me3 has distinct roles in the dynamic behaviors of the Cbx proteins and is important for the targeting of Cbx7 and Cbx8 to chromatin in mES cells, but plays a less important role for Cbx2, Cbx4, and Cbx6.

To further investigate the role of H3K27me3 in the targeting of the Cbx proteins to chromatin, we integrated *HaloTag-Cbx* fusion genes into the genome of $Ezh2^{-/-}$ mES cells. Ezh2 is the catalytic subunit of PRC2 (*Margueron and Reinberg, 2011*). The level of H3K27m3 was greatly reduced in $Ezh2^{-/-}$ mES cells in comparison with wild-type mES cells (*Figure 2E*). The residual level of H3K27me3 was most likely contributed

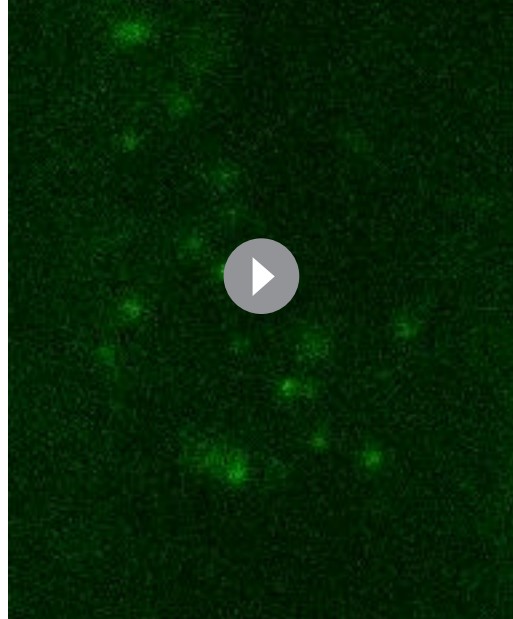

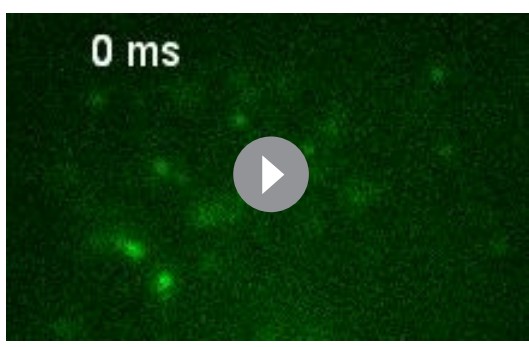

**Video 9.** HaloTag-Cbx2 in *Eed* KO mES cells (Fractional studies).

**Video 10.** HaloTag-Cbx4 in *Eed* KO mES cells (Fractional studies).

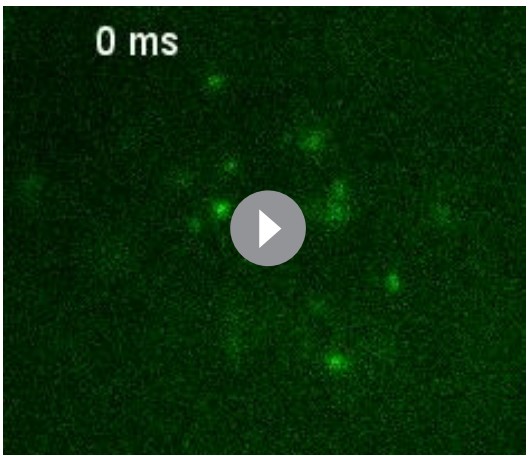

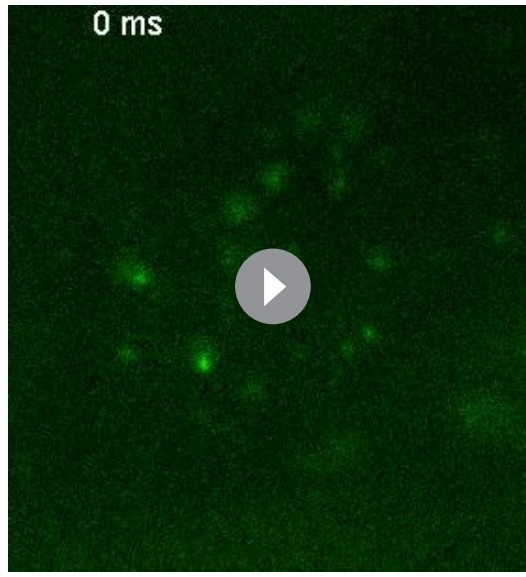

**Video 11.** HaloTag-Cbx6 in *Eed* KO mES cells (Fractional studies).

**Video 12.** HaloTag-Cbx7 in *Eed* KO mES cells (Fractional studies)

by Ezh1 (*Margueron et al., 2008*; *Shen et al., 2008*). We performed SMT of HaloTag-Cbx proteins in *Ezh2*$^{-/-}$ mES cells (*Video 14–18*). The CB levels of HaloTag-Cbx2, HaloTag-Cbx4, and HaloTag-Cbx6 in *Ezh2*$^{-/-}$ mES cells were similar to that in wild-type mES cells (*Figure 2D*, *Figure 2—figure supplement 1B*, *Supplementary file 1*, and *Figure 2—source data 1*). The fractional sizes of the CB components for HaloTag-Cbx7 and HaloTag-Cbx8 in *Ezh2*$^{-/-}$ mES cells were greatly reduced, in comparison with that in wild-type mES cells (*Figure 2B and D*, *Supplementary file 1*, and *Figure 2—source data 1*). Thus, our data further suggest that H3K27me3 is important for the targeting of Cbx7 and Cbx8 to chromatin, but plays a less important role for Cbx2, Cbx4, and Cbx6.

To dissect whether H3K27me3 has a direct role in the targeting of Cbx7 and Cbx8 to chromatin, we integrated *HaloTag-Cbx7/YFP-Eed* and *HaloTag-Cbx8/YFP-Eed* into the genome of *Eed*$^{-/-}$ mES cells and performed SMT of HaloTag-Cbx7 and HaloTag-Cbx8 (*Video 19* and *20*). The introduction of *YFP-Eed* fusion gene into *Eed*$^{-/-}$ mES cells restored the defective H3K27me3 level (*Figure 2E*). We measured $F_1 = (29 \pm 1)\%$ for HaloTag-Cbx7 and $F_1 = (27 \pm 1)\%$ for HaloTag-Cbx8, both of

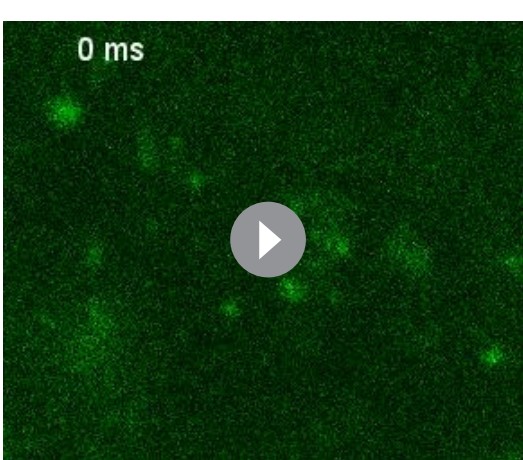

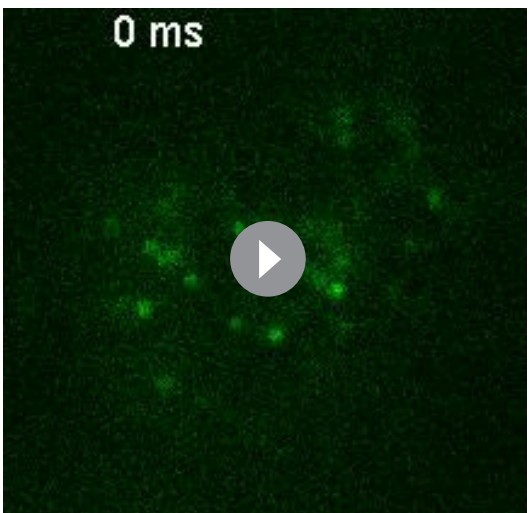

**Video 13.** HaloTag-Cbx8 in *Eed* KO mES cells (Fractional studies).

**Video 14.** HaloTag-Cbx2 in *Ezh2* KO mES cells (Fractional studies).

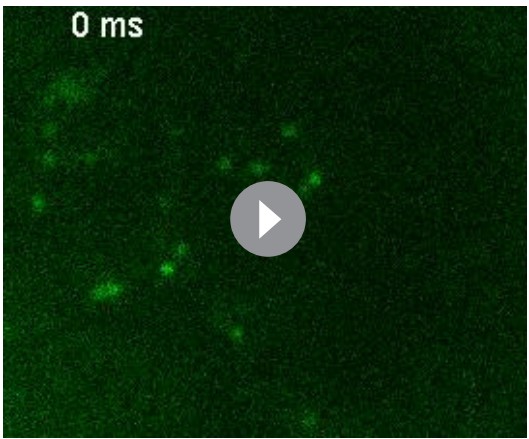

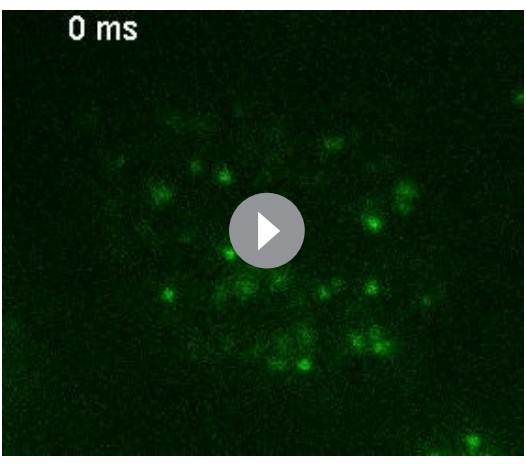

**Video 15.** HaloTag-Cbx4 in *Ezh2* KO mES cells (Fractional studies).

**Video 16.** HaloTag-Cbx6 in *Ezh2* KO mES cells (Fractional studies).

which are comparable to those obtained from wild-type mES cells (*Figure 2C and D*, *Figure 2—figure supplement 1C*, *Supplementary file 1*, and *Figure 2—source data 1*). Next, we integrated *HaloTag-Cbx7/YFP-Ezh2* and *HaloTag-Cbx8/YFP-Ezh2* into the genome of *Ezh2*$^{-/-}$ mES cells. The introduction of *YFP-Ezh2* into *Ezh2*$^{-/-}$ mES cells restored both the defective H3K27me3 level and the defective CB levels of HaloTag-Cbx7 and HaloTag-Cbx8 (*Figure 2C–2E*, *Figure 2—figure supplement 1C*, *Supplementary file 1*, *Video 21* and *22*, and *Figure 2—source data 1*). Altogether, our results demonstrate that H3K27me3 contributes significantly to the targeting of Cbx7 and Cbx8 to chromatin in mES cells.

## The Cbx7 CD contributes to, but is not efficient for the targeting of Cbx7 to chromatin

Our data indicate that among the Cbx family members, H3K27me3 is important for the targeting of Cbx7 and Cbx8 to chromatin, which seemingly does not reconcile with *in vitro* kinetic data where the Cbx7 CD (CD$_{Cbx7}$) has preference for H3K9me3 and the Cbx8 CD (CD$_{Cbx8}$) exhibit a weak affinity for both H3K27me3 and H3K9me3 (*Bernstein et al., 2006*; *Kaustov et al., 2011*; *Tardat et al.,*

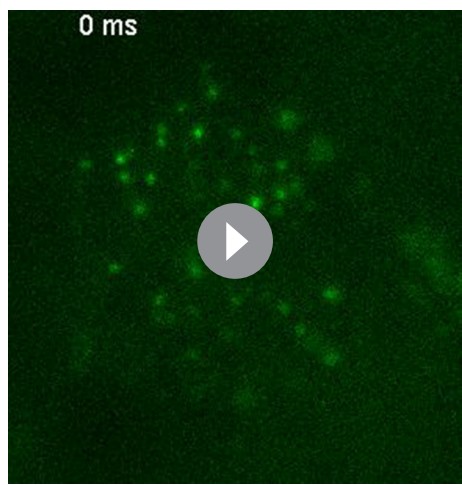

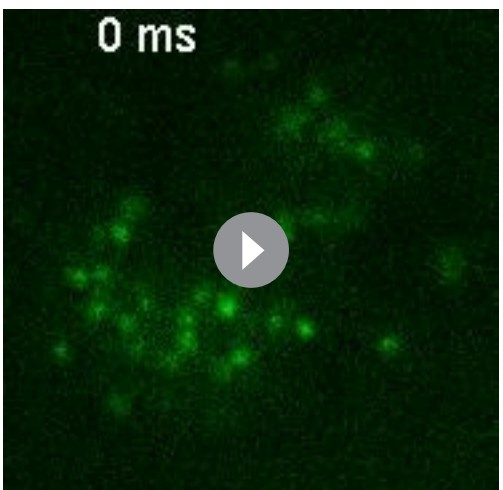

**Video 17.** HaloTag-Cbx7 in *Ezh2* KO mES cells (Fractional studies).

**Video 18.** HaloTag-Cbx8 in *Ezh2* KO mES cells (Fractional studies).

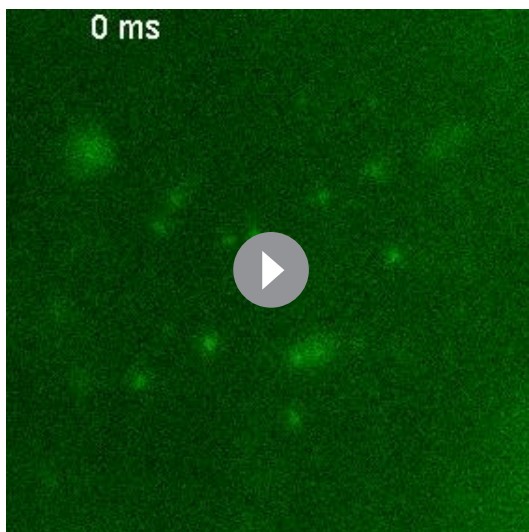

**Video 19.** HaloTag-Cbx7/Y-Eed in *Eed* KO mES cells (Fractional studies).

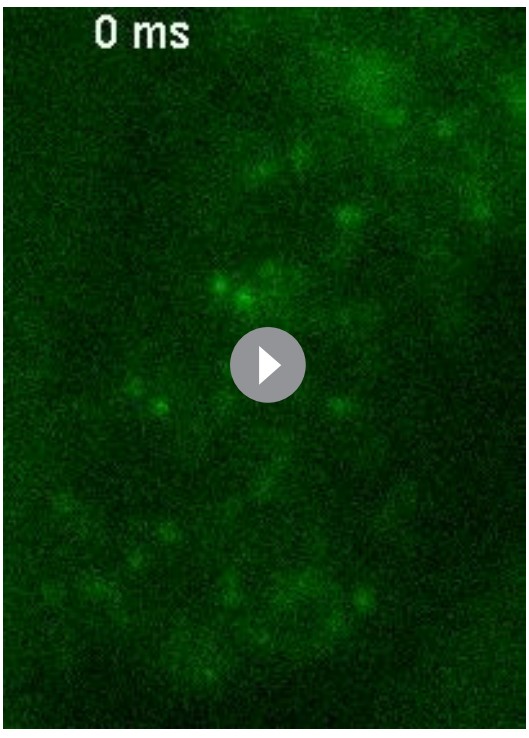

**Video 20.** HaloTag-Cbx8/Y-Eed in *Eed* KO mES cells (Fractional studies).

*2015*). In the following studies, we focused on the Cbx7 protein since (1) Cbx7 is smaller than Cbx8 (*Figure 1A* and *Figure 3A*); (2) Cbx7 contains three conserved domains while Cbx8 has four (*Figure 1A* and *Figure 3A*); (3) Cbx7-PRC1 is the major canonical PRC1 in mES cells (*Morey et al., 2013*; *Morey et al., 2012*); and (4) the expression of Cbx8 is nearly undetectable in mES cells (*Morey et al., 2013*; *Morey et al., 2012*). To test whether $CD_{Cbx7}$ binds to chromatin in living mES cells, we generated $CD_{Cbx7}$ fused with HaloTag (*Figure 3A*). The fusion was stably expressed in wild-type mES cells. We measured $F_1 = (8 \pm 1)\%$, $F_2 = (54 \pm 2)\%$, and $F_3 = (38 \pm 2)\%$ (*Figure 3B and C*, *Video 23*, *Supplementary file 1*, and *Figure 3—source data 1*). These data indicated that the fractional size of the CB component of HaloTag-$CD_{Cbx7}$ is below 30% of HaloTag-Cbx7,

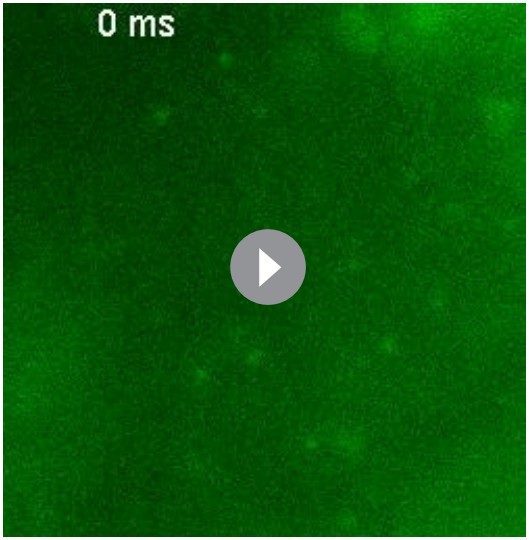

**Video 21.** HaloTag-Cbx7/Y-Ezh2 in *Ezh2* KO mES cells (Fractional studies).

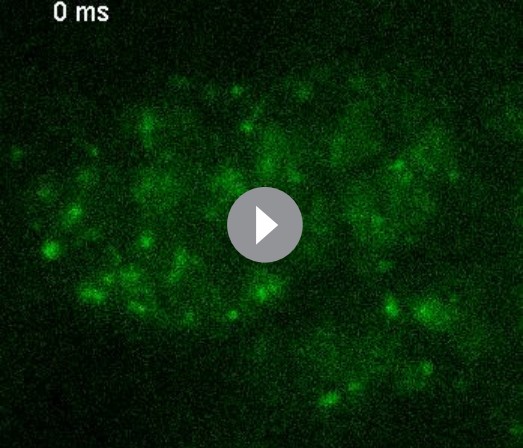

**Video 22.** HaloTag-Cbx8/Y-Ezh2 in *Ezh2* KO mES cells (Fractional studies).

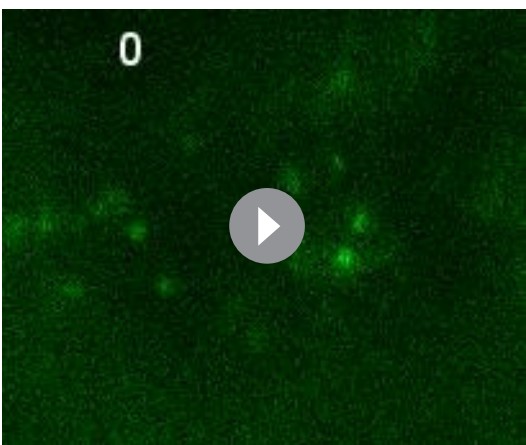

**Video 23.** HaloTag-CD$_{Cbx7}$ in wild-type mES cells (Fractional studies).

suggesting that CD$_{Cbx7}$ binds to chromatin less efficiently than Cbx7. Next, we generated Cbx7$^{F11A}$ and Cbx7$^{\triangle CD}$ fused with HaloTag, respectively (**Figure 3A**). The F11A mutation has been shown to disrupt the interaction of CD$_{Cbx7}$ and H3K27me3 (**Kaustov et al., 2011**). The two fusions were stably and correctly expressed in wild-type mES cells (**Figure 3—figure supplement 1**). We measured $F_1$ = (17 ± 1)%, $F_2$ = (42 ± 3)%, and $F_3$ = (41 ± 4)% for HaloTag-Cbx7$^{F11A}$, and $F_1$ = (13 ± 1)%, $F_2$ = (44 ± 1)%, and $F_3$ = (43 ± 2)% for Cbx7$^{\triangle CD}$ (**Figure 3B and C**, **Video 24 and 25**, **Supplementary file 1**, and **Figure 3— source data 1**), indicating that the CB fractional levels of Cbx7$^{F11A}$ and Cbx7$^{\triangle CD}$ are less than that of HaloTag-Cbx7, but more than that of CD$_{Cbx7}$. Taken together, our results imply that CD$_{Cbx7}$ is required, but not efficient for the targeting of Cbx7 to chromatin and that additional targeting mechanism(s) exist(s).

## Effects of the Cbx7 CD on the residence time of Cbx7 at chromatin

To determine the binding kinetics, we measured the *in vivo* residence time of Cbx7 molecules bound to chromatin. To reduce the photobleaching of JF$_{549}$, we performed time-lapse experiments at an integration time, $\tau_{int}$, of 30 ms interspersed with a dark time, $\tau_d$, of 170 ms (**Figure 4A** and **Video 26**). We calculated diffusion coefficients of individual HaloTag-Cbx7 molecules and considered molecules to be bound to chromatin if their $D_m$ was < 0.10 μm$^2$/s. The dwell times of individual stationary Cbx7 molecules were directly measured as the lifetime of the fluorescence spots. The cumulative frequency distributions of dwell times were fitted with a two-component exponential decay function (see Materials and methods) (**Figure 4B** and **Figure 4—source data 1**), generating two populations: the transient chromatin-bound population ($F_{1tb}$ and $\tau_{tb}$) and the stable chromatin-bound population ($F_{1sb}$ and $\tau_{sb}$). We measured $F_{1tb}$ = (23.4 ± 1.1)% ($\tau_{tb}$ = 0.79 ± 0.01 s) and $F_{1sb}$ = (5.3 ± 0.4)% ($\tau_{sb}$ = 7.3 ± 0.1 s) (**Figure 4C** and 4D, **Figure 4—figure supplement 1**, **Supplementary file 2**, and **Figure 4—source data 1**). We then investigated the residence times of the Cbx7 variants at chromatin (**Video 27–29**). The residence times of the stable chromatin-bound populations were determined for HaloTag-CD$_{Cbx7}$ ($\tau_{sb}$ = 4.7 ± 0.1 s), HaloTag-Cbx7$^{F11A}$ ($\tau_{sb}$ = 5.8 ± 0.1 s), and HaloTag-Cbx7$^{\triangle CD}$ ($\tau_{sb}$ = 4.7 ± 0.1 s) (**Figure 4C** and **Supplementary file 2**). Thus, the residence times of the stable chromatin-bound molecules of HaloTag-Cbx7$^{F11A}$ was longer than that of HaloTag-CD$_{Cbx7}$ and HaloTag-Cbx7$^{\triangle CD}$, but shorter than that of HaloTag-Cbx7 (**Figure 4C** and **Supplementary file 2**). The $F_{1sb}$ level of HaloTag-Cbx7$^{F11A}$ was higher than that of HaloTag-CD$_{Cbx7}$ and HaloTag-Cbx7$^{\triangle CD}$, but was less than that of HaloTag-Cbx7 (**Figure 4D** and **Supplementary file 2**). To allow visual comparison among Cbx7 and its variants, we plotted their survival probability in the same figure (**Figure 4E** and **Figure 4—source data 1**). We observed that HaloTag-Cbx7$^{F11A}$ stays a longer time at chromatin than HaloTag-CD$_{Cbx7}$ and HaloTag-Cbx7$^{\triangle CD}$, but a shorter time than HaloTag-Cbx7 (**Figure 4E**). Altogether, our results suggest that the interaction of H3K27me3 and CD$_{Cbx7}$ is not enough for the stabilizing of Cbx7 at chromatin.

## Disruption of the complex formation of Cbx7-PRC1 facilitates the targeting of Cbx7 to chromatin

To search for factor(s) that contribute(s) to the targeting of Cbx7 to chromatin, we investigated whether individual components of Cbx7-PRC1 affect the targeting of Cbx7 to chromatin. We integrated *HaloTag-Cbx7* into the genome of *Ring1a$^{-/-}$/Ring1b$^{-/-}$* mES cells. Ring1a and Ring1b are the assemblage of PRC1 complexes and the depletion of *Ring1b* disrupts the complex formation of PRC1 (**Leeb and Wutz, 2007**). The *logD$_m$* histograms indicated three components (**Figure 5A**, **Video 30**, and **Figure 5—source data 1**). We measured $F_1$ = (44 ± 1)%, $F_2$ = (32 ± 2)%, and $F_3$ = (24

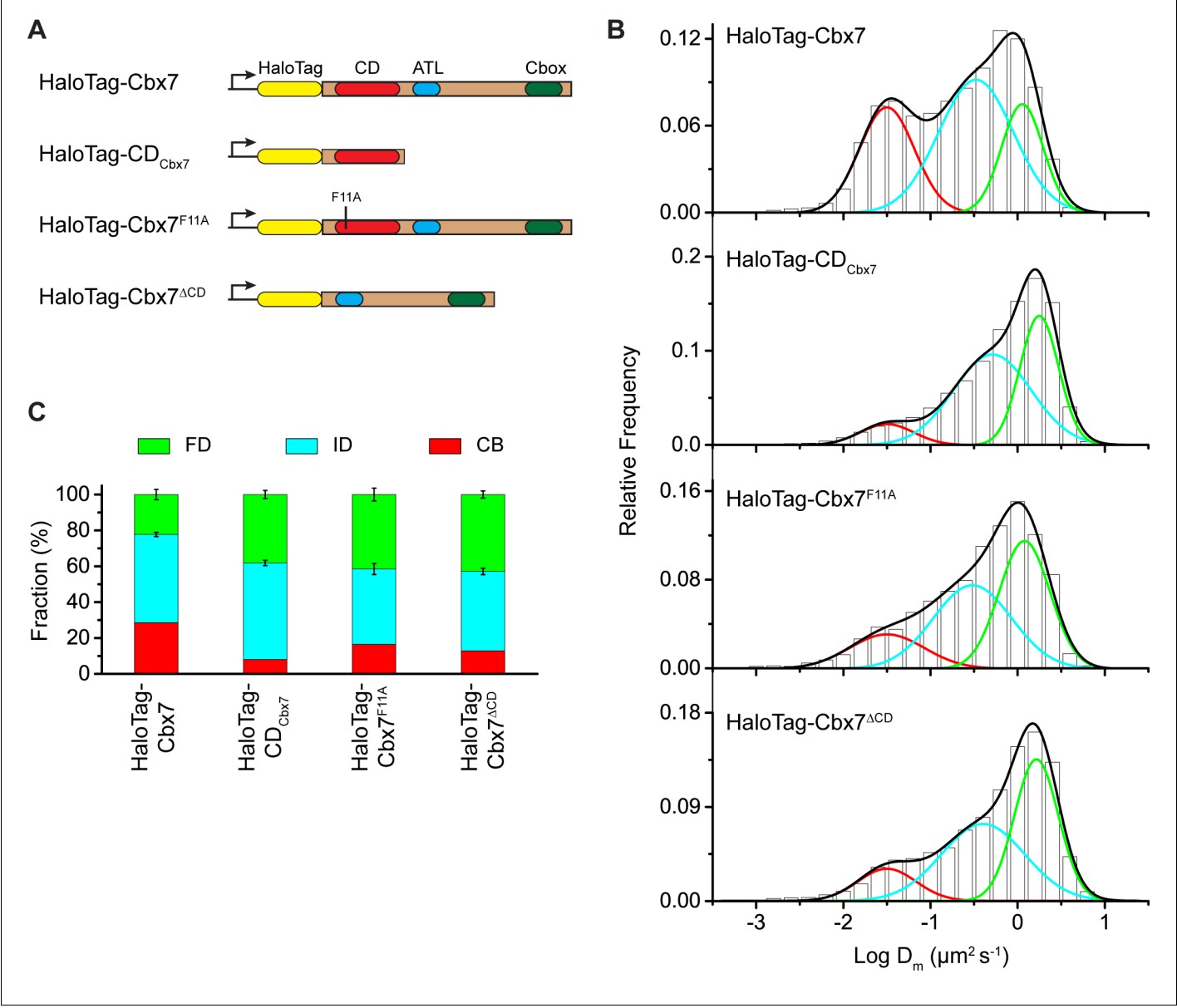

**Figure 3.** The Cbx7 CD is not efficient for the targeting of Cbx7 to chromatin. (**A**) Schematic representation of Cbx7 variants. (**B**) Normalized histograms of the log maximum likelihood diffusion coefficient $D_m$ for HaloTag-Cbx7 replicated from *Figure 1E* and for HaloTag-CD$_{Cbx7}$ (N = 24 cells, n = 6600 trajectories), HaloTag-Cbx7$^{F11A}$ (N = 22 cells, n = 1882 trajectories), and HaloTag-Cbx7$^{\triangle CD}$ (N = 15 cells, n = 5215 trajectories) in wild-type mES cells. The histograms of HaloTag-CD$_{Cbx7}$, HaloTag-Cbx7$^{F11A}$, and HaloTag-Cbx7$^{\triangle CD}$ were fitted with a three-component Gaussian. (**C**) Fraction of the CB, ID, and FD population for HaloTag-Cbx7 replicated from *Figure 1F*, HaloTag-CD$_{Cbx7}$, HaloTag-Cbx7$^{F11A}$, and HaloTag-Cbx7$^{\triangle CD}$. The data were obtained from *Figure 3B* fitted with a Gaussian. Results are means ± SD.

The following source data and figure supplement are available for figure 3:

**Source data 1.** Source data for *Figure 3B*
**Figure supplement 1.** Control experiments for analyzing the protein levels of the HaloTag-Cbx7 variants.

± 2)% (*Figure 5B* and *Supplementary file 1*), indicating that the fractional size of the CB component is ~1.5 fold of that obtained from wild-type mES cells. Ring1a and Ring1b are not only core components of Cbx-PRC1, but also core components of vPRC1 (*Gao et al., 2012*; *Tavares et al., 2012*). To

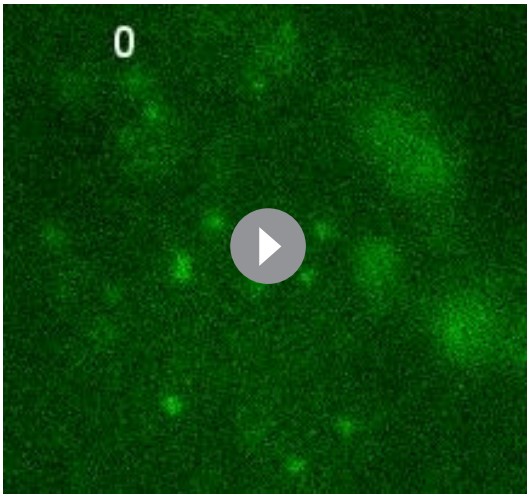

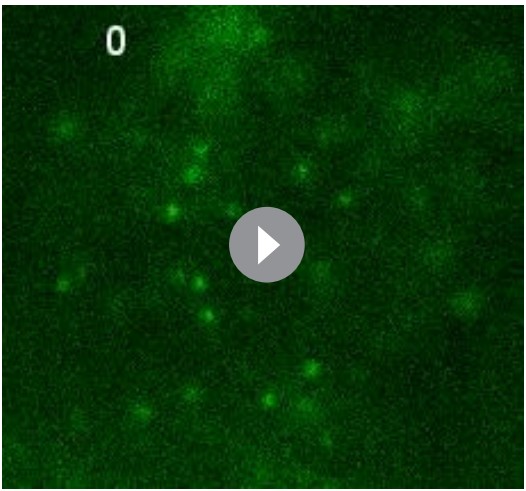

**Video 24.** HaloTag-Cbx7$^{F11A}$ in wild-type mES cells (Fractional studies).

**Video 25.** HaloTag-Cbx7$^{\triangle CD}$ in wild-type mES cells (Fractional studies).

disrupt the complex formation of Cbx-PRC1, we established $Bmi1^{-/-}/Mel18^{-/-}$ mES cells stably expressing *HaloTag-Cbx7*. Bmi1 and Mel18 are assembled into the Cbx-PRC1 complexes (*Gao et al., 2012*). We performed SMT of HaloTag-Cbx7 in $Bmi1^{-/-}/Mel18^{-/-}$ mES cells (*Video 31*). The $logD_m$ histograms indicated three populations (*Figure 5A* and *Figure 5—source data–1*). We measured $F_1 = (40 \pm 1)\%$, $F_2 = (44 \pm 1)\%$, and $F_3 = (16 \pm 5)\%$ (*Figure 5B* and *Supplementary file 1*), indicating that the fractional size of the CB component is ~1.4 fold of that obtained from wild-type mES cells. Thus, these data suggest that the disruption of the complex formation of Cbx7-PRC1 facilitates the targeting of Cbx7 to chromatin.

To investigate whether the increased level of Cbx7 at chromatin upon the depletion of PRC1 components is caused by the dissociation rate, we measured the residence times of HaloTag-Cbx7 in $Ring1a^{-/-}/Ring1b^{-/-}$ and $Bmi1^{-/-}/Mel18^{-/-}$ mES cells, respectively (*Video 32* and *33*). The cumulative frequency distributions of dwell times were fitted with a two-component exponential decay model (*Figure 5—figure supplement 1*, and *Figure 5—source data 1*)). We measured $\tau_{tb} = 1.06 \pm 0.02$ s and $\tau_{sb} = 10.6 \pm 0.2$ s for HaloTag-Cbx7 in $Ring1a^{-/-}/Ring1b^{-/-}$ mES cells and $\tau_{tb} = 1.08 \pm 0.02$ s and $\tau_{sb} = 10.9 \pm 0.1$ s for HaloTag-Cbx7 in $Bmi1^{-/-}/Mel18^{-/-}$ mES cells (*Figure 5C* and *Supplementary file 2*), indicating that HaloTag-Cbx7 stays a longer time at chromatin in $Ring1a^{-/-}/Ring1b^{-/-}$ and $Bmi1^{-/-}/Mel18^{-/-}$ mES cells than in wild-type mES cells. The $F_{1tb}$ and $F_{1sb}$ levels for HaloTag-Cbx7 in $Ring1a^{-/-}/Ring1b^{-/-}$ and $Bmi1^{-/-}/Mel18^{-/-}$ mES cells were higher than that in wild-type mES cells (*Figure 5D* and *Supplementary file 2*). A survival probability plot showed that HaloTag-Cbx7 stays a shorter time at chromatin in wild-type mES cells than in $Ring1a^{-/-}/Ring1b^{-/-}$ and $Bmi1^{-/-}/Mel18^{-/-}$ mES cells (*Figure 5E* and *Figure 5—source data 1*). Thus, our data demonstrate that genetic depletion of PRC1 components facilitates the stabilizing of Cbx7 at chromatin.

## CD$_{Cbx7}$ and ATL$_{Cbx7}$ together constitute a DNA-binding unit

Since the Cbx-PRC1 components tested are not required for the targeting of Cbx7 to chromatin, we turned our attention into Cbx7 itself. In addition to the conserved CD, Cbx7 harbors an ATL motif adjacent to CD (*Senthilkumar and Mishra, 2009*) (*Figure 6A*). Since the Cbx7 ATL (ATL$_{Cbx7}$) contains 6 basic amino acids out of 16, we postulated that the ATL motif may be involved in nucleic acid-binding. To test this hypothesis, we generated Cbx7 variants (*Figure 6—figure supplement 1*) and performed electrophoretic mobility shift assay (EMSA) (*Figure 6B*). EMSA analysis indicated that CD$_{Cbx7}$ has no DNA-binding activity, consistent with early studies (*Bernstein et al., 2006*). Although ATL$_{Cbx7}$ contains a high content of basic amino acids, EMSA analysis demonstrated that ATL$_{Cbx7}$ has undetectable DNA-binding activity, consistent with previous report that ATL does not bind to DNA (*Reeves and Nissen, 1990*). However, under the same conditions, CD-ATL$_{Cbx7}$ showed clear DNA-

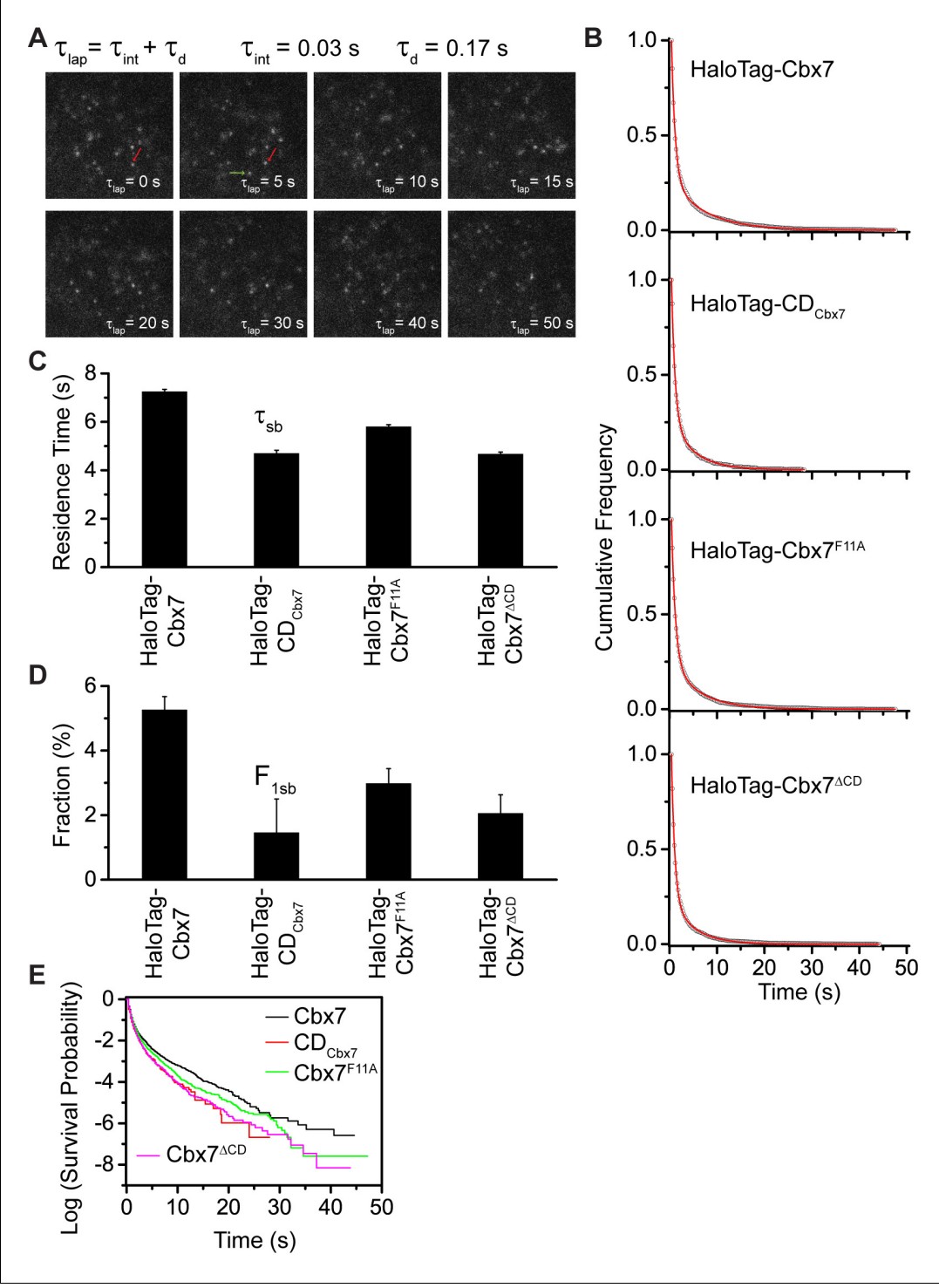

**Figure 4.** Effects of the Cbx7 CD on the residence time of Cbx7 at chromatin. (**A**) Time-lapse imaging of HaloTag-Cbx7 at constant integration ($\tau_{int}$ = 30 ms) and dark ($\tau_d$ = 170 ms) time in wild-type mES cells. The red arrow indicates a molecule that binds to chromatin. The green arrow indicates a diffusing molecule. Molecules with $D_m <$ 0.10 μm²/s were selected to calculate residence time and survival probability. (**B**) Cumulative frequency distribution of the dwell times for HaloTag-Cbx7 (N = 17 cells, n = 2169 trajectories), HaloTag-CD$_{Cbx7}$ (N = 18 cells, n = 790 trajectories), HaloTag-Cbx7$^{F11A}$ (N = 25 cells, n = 3956 trajectories), and HaloTag-Cbx7$^{\triangle CD}$ (N = 21 cells, n = 3471 trajectories) in wild-type mES cells. The histograms were fitted with a two-component exponential decay model. (**C**) Residence time ($\tau_{sb}$) of the stable chromatin-bound population for HaloTag-Cbx7, HaloTag-CD$_{Cbx7}$, HaloTag-

*Figure 4 continued on next page*

*Figure 4 continued*

Cbx7[F11A], and HaloTag-Cbx7$^{\triangle CD}$ in wild-type mES cells. (**D**) Fraction (F$_{1sb}$) of the stable chromatin-bound population for HaloTag-Cbx7, HaloTag-CD$_{Cbx7}$, HaloTag-Cbx7[F11A], and HaloTag-Cbx7$^{\triangle CD}$ in wild-type mES cells. (**E**) Survival probability for HaloTag-Cbx7, HaloTag-CD$_{Cbx7}$, HaloTag-Cbx7[F11A], and HaloTag-Cbx7$^{\triangle CD}$ in wild-type mES cells.

The following source data and figure supplement are available for figure 4:

**Source data 1.** Source data for *Figure 4B* and *Figure 4—figure supplementary 1*
**Figure supplement 1.** Control experiments for determine photobleaching constant of JF$_{549}$.

binding activity, suggesting that the DNA-binding activity requires both CD$_{Cbx7}$ and ATL$_{Cbx7}$. To test whether the basic amino acids of ATL$_{Cbx7}$ affect the DNA-binding activity, we substituted these basic amino acids with alanine or glycine to generate CD-ATLm$_{Cbx7}$. EMSA analysis showed that the substitution abolishes the DNA-binding activity of CD-ATL$_{Cbx7}$. As a control, GST did not bind to DNA. The DNA-binding capacity of CD-ATL$_{Cbx7}$ is concentration-dependent (*Figure 6C*). The K$_d$ was determined to be ~1.0 µM, which is much smaller than the CD$_{Cbx7}$ binding to H3K27me3 peptide (*Bernstein et al., 2006*; *Kaustov et al., 2011*; *Tardat et al., 2015*).

Previous studies have shown that CD$_{Cbx7}$ binds to RNA with affinity of ~50 µM (*Bernstein et al., 2006*; *Yap et al., 2010*). To compare the relative affinity of CD-ATL$_{Cbx7}$ binding to DNA versus RNA, we performed competitive assays. 0.5 µM of double-stranded DNA-2 (dsDNA-2), with sequence being different from dsDNA-1, completely dissociated the fluorescently labeled dsDNA-1 (0.1 µM) from CD-ATL$_{Cbx7}$ (*Figure 6D*). At 4.0 µM, single-stranded DNA (ssDNA) could not completely dissociate dsDNA-1 from CD-ATL$_{Cbx7}$ (*Figure 6E*). Under the identical conditions, 10 µM of double-stranded RNA could not completely dissociate dsDNA-1 from CD-ATL$_{Cbx7}$ (*Figure 6F*). Likewise, 40 µM of single-stranded RNA had no noticeable effects on the association CD-ATL$_{Cbx7}$ with dsDNA-1 (*Figure 6G*). Altogether, our results demonstrate that CD-ATL$_{Cbx7}$ preferentially recognizes dsDNA rather than ssDNA, dsRNA and ssRNA.

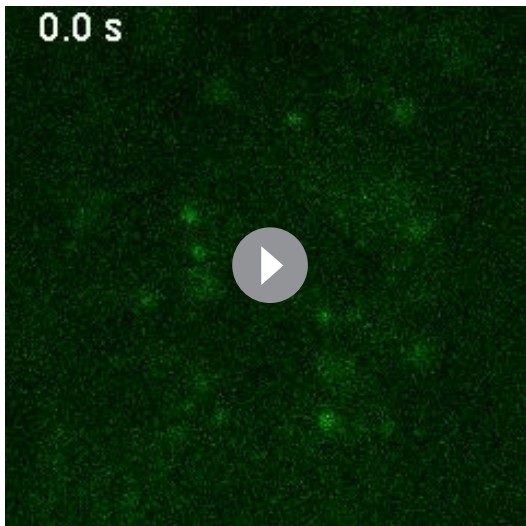

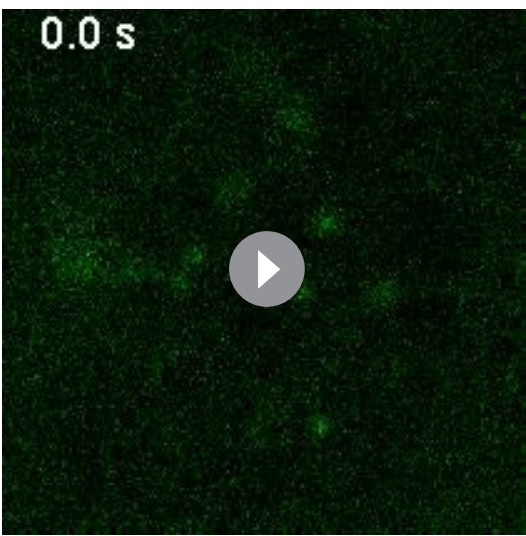

**Video 26.** HaloTag-Cbx7 in wild-type mES cells (Residence time studies).

**Video 27.** HaloTag-CD$_{Cbx7}$ in wild-type mES cells (Residence time studies).

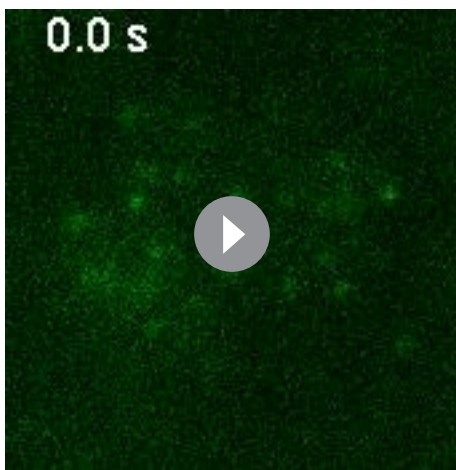

**Video 28.** HaloTag-Cbx7$^{F11A}$ in wild-type mES cells (Residence time studies).

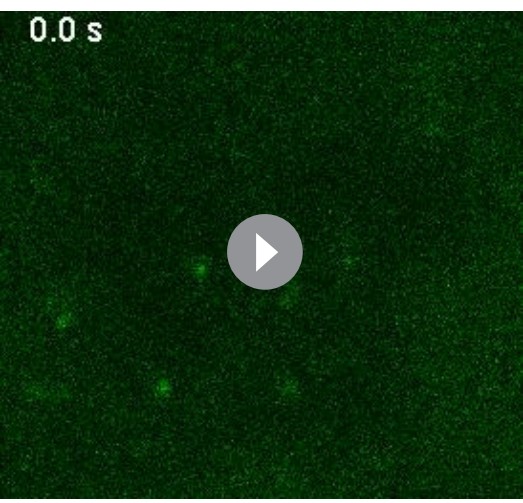

**Video 29.** HaloTag-Cbx7$^{\triangle CD}$ in wild-type mES cells (Residence time studies).

## CD$_{Cbx7}$ and ATL$_{Cbx7}$ together control the targeting of Cbx7 to chromatin

To investigate whether the *in vitro* capacity of the binding of CD-ATL$_{Cbx7}$ to DNA has a functional role in the targeting of Cbx7 to chromatin *in vivo*, we made Cbx7 variants lacking DNA-binding ability and stably expressed these variants in wild-type mES cells (*Figure 7A* and *Figure 3—figure supplement 1*). The histograms for HaloTag-Cbx7$^{\triangle ATL}$ were fitted with three populations (*Figure 7B*, *Video 34*, and *Figure 7—source data 1*). We measured F$_1$ = (16 ± 1)%, F$_2$ = (44 ± 2)%, and F$_3$ = (40 ± 3)% (*Figure 7C* and *Supplementary file 1*), indicating that the fractional size of the CB component of HaloTag-Cbx7$^{\triangle ATL}$ is about half of that of HaloTag-Cbx7. The histograms for HaloTag-Cbx7$^{ATLm}$ indicated three components (*Figure 7B*, *Video 35*, and *Figure 7—source data 1*). We measured F$_1$ = (16 ± 1)%, F$_2$ = (60 ± 1)%, and F$_3$ = (24 ± 3)% (*Figure 7C* and *Supplementary file 1*), indicating that the fractional size of the CB component of HaloTag-Cbx7$^{ATLm}$ is comparable to that of HaloTag-Cbx7$^{\triangle ATL}$. Thus, our data indicate that the ATL$_{Cbx7}$ motif is required for the efficient targeting of Cbx7 to chromatin.

Given that CD$_{Cbx7}$ and ATL$_{Cbx7}$ constitute an H3K27me3- and DNA-binding unit, we tested the role of CD-ATL$_{Cbx7}$ in the targeting of Cbx7 to chromatin. We deleted both CD$_{Cbx7}$ and ATL$_{Cbx7}$ to generate HaloTag-Cbx7$^{\triangle CD-ATL}$ that was stably expressed in wild-type mES cells (*Figure 7A* and *Figure 3—figure supplement 1*). The histograms for HaloTag-Cbx7$^{\triangle CD-ATL}$ were fitted with three populations (*Figure 7B*, *Video 36*, and *Figure 7—source data 1*). We measured F$_1$ = (8 ± 3)%, F$_2$ = (46 ± 1)%, and F$_3$ = (46 ± 1)% for HaloTag-Cbx7$^{\triangle CD-ATL}$ (*Figure 7C* and *Supplementary file 1*), indicating that the F$_1$ level was lower than that for HaloTag-Cbx7$^{\triangle CD}$ and HaloTag-Cbx7$^{\triangle ATL}$ (*Supplementary file 1*). Thus, our data suggest that CD$_{Cbx7}$ and ATL$_{Cbx7}$ function together as a unit to mediate the targeting of Cbx7 to chromatin.

Next, we investigated the residence times of the Cbx7 variants at chromatin (*Figure 7—figure supplement 1*, *Video 37–39*, and *Figure 7—source data 1*). The residence time of the stable chromatin-bound population was reduced for HaloTag-Cbx7$^{ALTm}$ ($\tau_{sb}$ = 6.0 ± 0.2 s), HaloTag-Cbx7$^{\triangle ATL}$ ($\tau_{sb}$ = 5.8 ± 0.1 s), and HaloTag-Cbx7$^{\triangle CD-ATL}$ ($\tau_{sb}$ = 4.6 ± 0.1 s), in comparison with HaloTag-Cbx7 ($\tau_{sb}$ = 7.3 ± 0.1 s) (*Figure 7D* and *Supplementary file 2*). The F$_{1sb}$ levels for HaloTag-Cbx7$^{ALTm}$ and HaloTag-Cbx7$^{\triangle ATL}$ were less than that for HaloTag-Cbx7 (*Figure 7E* and *Supplementary file 2*). The F$_{1sb}$ level for HaloTag-Cbx7$^{\triangle CD-ATL}$ was almost undetectable (*Figure 7E* and *Supplementary file 2*). A survival probability plot demonstrated that these HaloTag-Cbx7 variants stay a shorter time period at chromatin than HaloTag-Cbx7 (*Figure 7F* and *Figure 7—source data 1*). Thus, our data demonstrate that CD-ATL$_{Cbx7}$ is a functional unit that regulates the targeting of Cbx7 to chromatin.

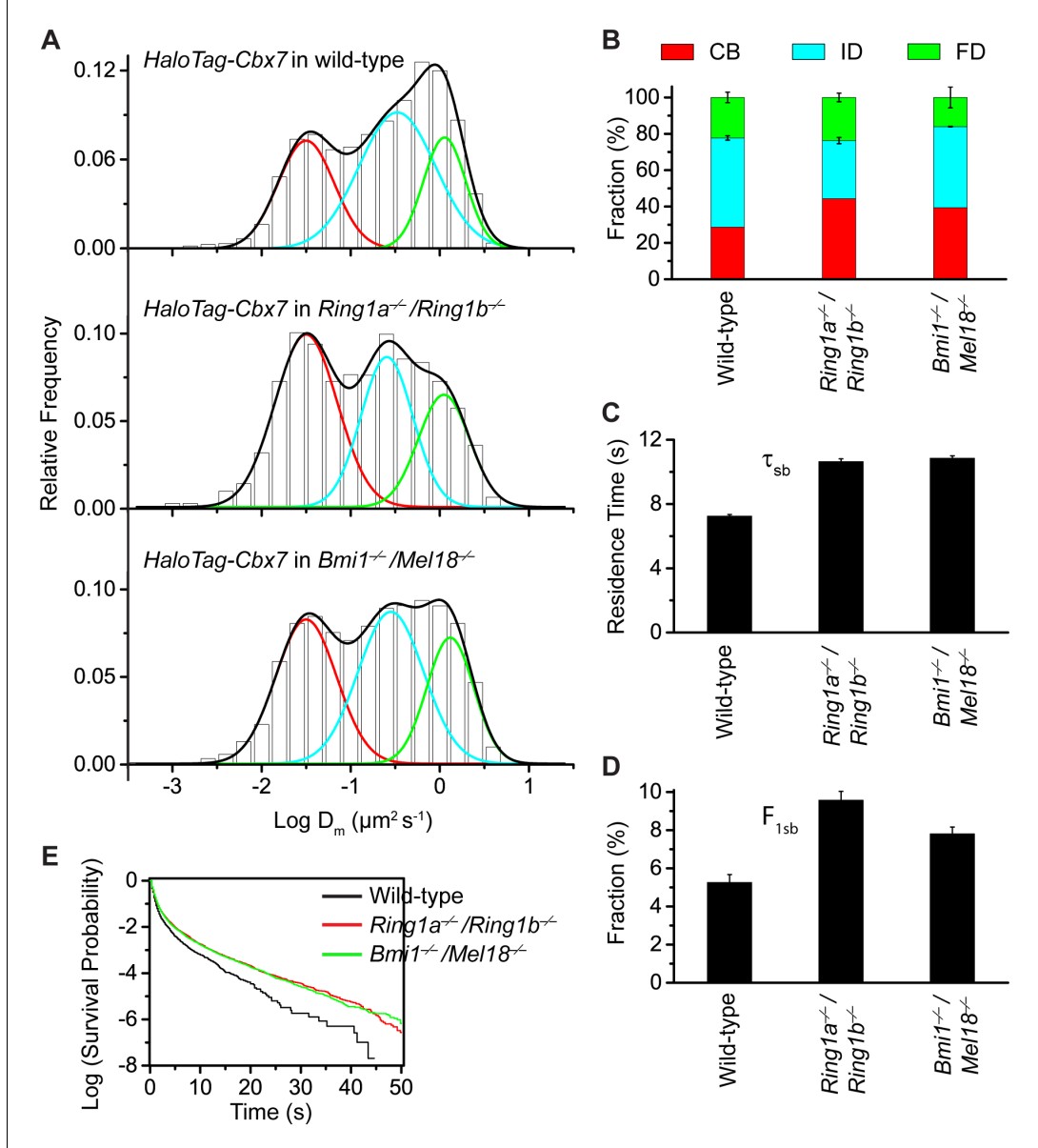

**Figure 5.** Disruption of the complex formation of Cbx7-PRC1 facilitates the targeting of Cbx7 to chromatin. (**A**) Normalized histograms of the log maximum likelihood diffusion coefficient $D_m$ for HaloTag-Cbx7 in wild-type mES cells replicated from *Figure 1E* and for HaloTag-Cbx7 in $Ring1a^{-/-}$ / $Ring1b^{-/-}$ (N = 29 cells, n = 2600 trajectories) and $Bmi1^{-/-}/Mel18^{-/-}$ (N = 27 cells, n = 6859 trajectories) mES cells. The histograms were fitted with a three-component Gaussian. (**B**) Fraction of the CB (red), ID (cyan), and FD (green) population for HaloTag-Cbx7 in wild-type mES cells replicated from *Figure 1F* and for HaloTag-Cbx7 in $Ring1a^{-/-}/Ring1b^{-/-}$ and $Bmi1^{-/-}/Mel18^{-/-}$ mES cells. Results are means ± SD. (**C–D**) Residence time (**C**) and fraction (**D**) of the stable chromatin-bound population for HaloTag-Cbx7 in wild-type mES cells replicated from *Figure 4C and D* and for HaloTag-Cbx7 in $Ring1a^{-/-}/Ring1b^{-/-}$ (N = 18 cells, n = 4849 trajectories) and $Bmi1^{-/-}/Mel18^{-/-}$ (N = 27 cells, n = 3484 trajectories) mES cells. (**E**) Survival probability for HaloTag-Cbx7 in wild-type mES cells replicated from *Figure 4E*, and for HaloTag-Cbx7 in $Ring1a^{-/-}/Ring1b^{-/-}$ and $Bmi1^{-/-}/Mel18^{-/-}$ mES cells.

The following source data and figure supplement are available for figure 5:

**Source data 1.** Source data for *Figure 5A* and *Figure 5—figure supplementary 1*

**Figure supplement 1.** Cumulative frequency distribution of the dwell times for determining the residence times of HaloTag-Cbx7 in $Ring1a^{-/-}$ / $Ring1b^{-/-}$ and $Bmi1^{-/-}/Mel18^{-/-}$ mES cells.

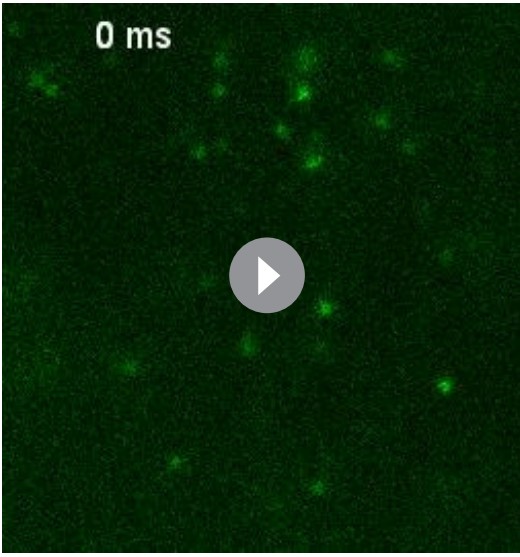

**Video 30.** HaloTag-Cbx7 in *Ring1a/Ring1b* dKO mES cells (Fractional studies).

**Video 31.** HaloTag-Cbx7 in *Bmi1/Mel18* dKO mES cells (Fractional studies).

## Discussion

In this study, we have elucidated the recruitment mechanism for the Cbx-PRC1 complexes by integrating approaches from live-cell SMT, genetic engineering, and biochemistry. We have demonstrated that H3K27me3 has a central and a direct role in the recruitment of Cbx7 and Cbx8 to chromatin *in vivo*, while plays a less important role in the targeting of Cbx2, Cbx4, and Cbx6 to chromatin. We have identified that the CD-ATL$_{Cbx7}$ cassette functions as a unit that co-recognizes H3K27me3 and DNA and regulates the targeting of Cbx7 to chromatin. These results challenge the prevailing view that all Cbx family members require H3K27me3 for the targeting of them to chromatin and provide new insights into the genetic, biochemical, and genome-wide analysis for our understanding of the Cbx-PRC1 targeting mechanisms. We propose that a hierarchical cooperation between a low-affinity H3K27me3-binding CD$_{Cbx7}$ and a high-affinity DNA-binding CD-ATL$_{Cbx7}$ targets Cbx7-PRC1 to chromatin.

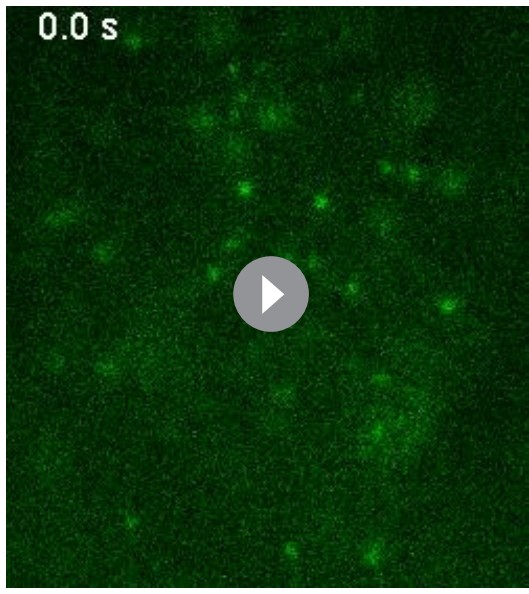

**Video 32.** HaloTag-Cbx7 in *Ring1a/Ring1b* dKO mES cells (Residence time studies).

**Video 33.** HaloTag-Cbx7 in *Bmi1/Mel18* dKO mES cells (Residence time studies).

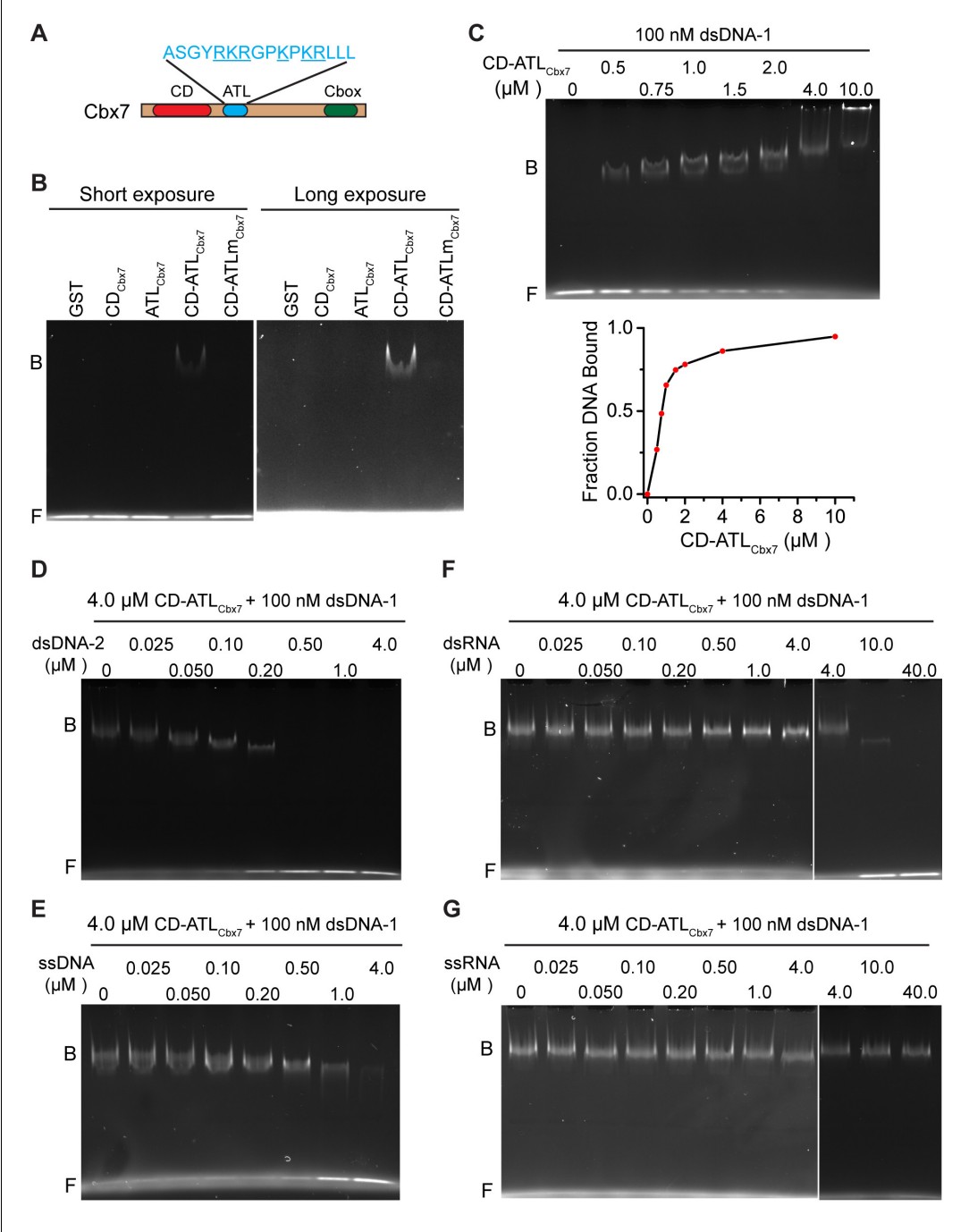

**Figure 6.** $CD_{Cbx7}$ and $ATL_{Cbx7}$ together constitute a DNA-binding entity. (**A**) Schematic representation of Cbx7. The sequence of amino acids of ATL motif is shown. The basic amino acids are underlined and mutated to alanine to generate ATLm. (**B**) EMSA for the determination of Cbx7 variants binding to dsDNA-1. dsDNA-1 was labelled with Alexa Fluor 488 dye. Left: short-time exposure. Right: long-time exposure. B: bound DNA-protein complex. F: free DNA. (**C**) EMSA for the determination of the dissociation constant ($K_d$) of the $CD-ATL_{Cbx7}$ cassette binding to dsDNA-1. Bottom: binding curve for the $CD-ATL_{Cbx7}$ cassette. (**D–G**) EMSA for the determination of the relative affinities for the $CD-ATL_{Cbx7}$ cassette binding to dsDNA-2, ssDNA, dsRNA, and ssRNA. dsDNA-1 within dsDNA-1/$CD-ATL_{Cbx7}$ complexes was competed with competitors, dsDNA-2 (**D**), ssDNA (**E**), dsRNA (**F**), and ssRNA (**G**), respectively. ds: double-strand. ss: single-strand.

The following figure supplement is available for figure 6:

**Figure supplement 1.** Control experiments for analyzing the Cbx7 variants purified from BL21 cells by SDS-PAGE.

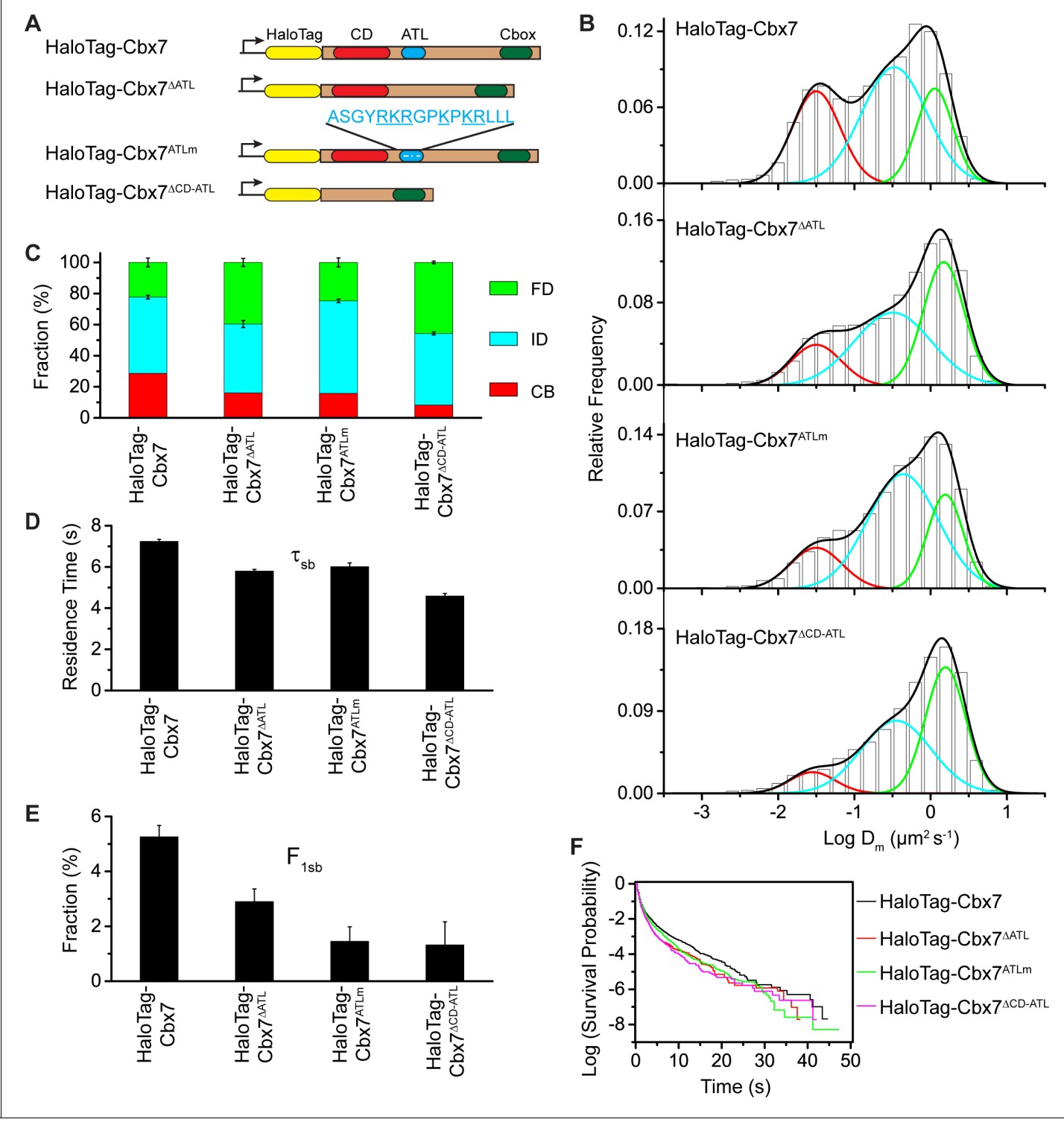

**Figure 7.** CD$_{Cbx7}$ and ATL$_{Cbx7}$ together control the targeting of Cbx7 to chromatin. (A) Schematic representation of Cbx7 variants. The underlined ATL amino acids were mutated into alanine or glycine. (B) Normalized histograms of the log maximum likelihood diffusion coefficient $D_m$ for HaloTag-Cbx7 replicated from **Figure 1E** and for HaloTag-Cbx7$^{\triangle ATL}$ (N = 12 cells, n = 3065 trajectories), HaloTag-Cbx7$^{ATLm}$ (N = 13 cells, n = 2257 trajectories), and HaloTag-Cbx7$^{\triangle CD-ATL}$ (N = 35 cells, n = 8329 trajectories) in wild-type mES cells. The histograms were fitted with a three-component Gaussian. (C) Fraction of the CB, ID, and FD population for HaloTag-Cbx7 replicated from **Figure 1F**, HaloTag-Cbx7$^{\triangle ATL}$, HaloTag-Cbx7$^{ATLm}$, and HaloTag-Cbx7$^{\triangle CD-ATL}$. The data were obtained from **Figure 7B** fitted with a Gaussian. Results are means ± SD. (D–E) Residence time (D) and fraction (E) of the stable chromatin-bound population for HaloTag-Cbx7 replicated from **Figure 4C and D**, HaloTag-Cbx7$^{\triangle ATL}$ (N = 17 cells, n = 2384 trajectories),

*Figure 7 continued on next page*

*Figure 7 continued*

HaloTag-Cbx7[ATLm] (N = 24 cells, n = 2957 trajectories), and HaloTag-Cbx7[△CD-ATL] (N = 22 cells, n = 4908 trajectories). Results are means ± SD. (**F**) Survival probability for HaloTag-Cbx7 replicated from *Figure 4D*, HaloTag-Cbx7[△ATL], HaloTag-Cbx7[ATLm], and HaloTag-Cbx7[△CD-ATL].

The following source data and figure supplement are available for figure 7:

**Source data 1.** Source data for *Figure 7B* and *Figure 7—figure supplementary 1*.

**Figure supplement 1.** Cumulative frequency distribution of the dwell times for determining the residence times of HaloTag-Cbx7[△ATL], HaloTag-Cbx7[ATLm], and HaloTag-Cbx7[△CD-ATL] in wild-type mES cells.

## Targeting the Cbx family members with and without dependence on PRC2 and H3K27me3

During evolution, the number of genes encoding Cbx proteins has increased, which has resulted in structural and functional diversification (*Cheng et al., 2014*; *Klauke et al., 2013*; *Morey et al., 2012*; *Ren and Kerppola, 2011*; *Ren et al., 2008*; *Tatavosian et al., 2015*; *Vincenz and Kerppola, 2008*; *Whitcomb et al., 2007*; *Zhen et al., 2014*). At the single-molecule level, we quantified the kinetic fractions of the Cbx proteins within living mES cells and revealed that ~ 30% of Cbx2, Cbx7, and Cbx8 associate with chromatin at a given time period while ~10–15% of Cbx4 and Cbx6 bind to chromatin. The fractional sizes and diffusion constants of the ID populations among the Cbx family members are distinct, suggesting that the Cbx proteins employ distinct mechanisms to explore the nucleus of the cell.

At the single-molecule sensitivity, we demonstrated that Cbx7 and Cbx8 are displaced from chromatin in *Eed*[−/−] and *Ezh2*[−/−] mES cells. The introduction of *Eed* into *Eed*[−/−] mES cells and of *Ezh2* into *Ezh2*[−/−] mES cells restored the Cbx7 and Cbx8 association with chromatin and the H3K27me3 level. Thus, it is likely that H3K27me3 directly controls the association of Cbx7 and Cbx8 with chromatin. Consistent with this notion, previous genome-wide ChIP-Seq analysis demonstrated that Cbx7 is displaced from chromatin in *Eed*[−/−] mES cells (*Morey et al., 2013*). We found that the removal of H3K27me3 has no or small effects on the association of Cbx2, Cbx4, and Cbx6 with chromatin. No effects on the Cbx6 association with chromatin is consistent with previous studies where Cbx6 does not interact with Ring1b and only 5% of Cbx6 target genes are occupied by H3K27me3

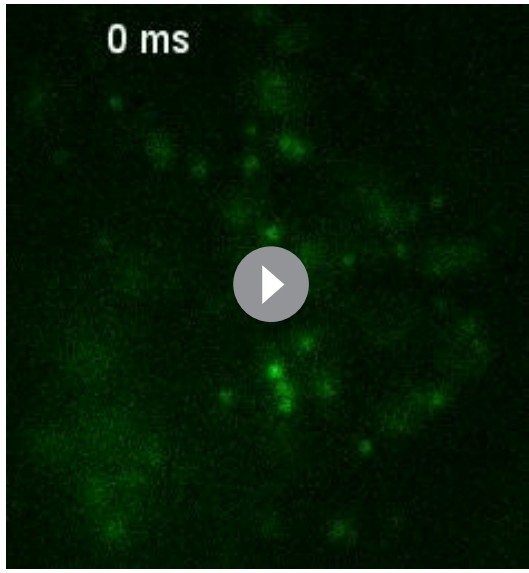

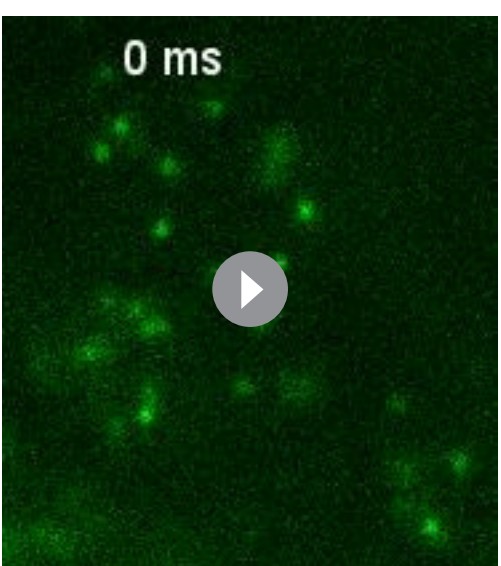

**Video 34.** HaloTag-Cbx7[△ATL] in wild-type mES cells (Fractional studies).

**Video 35.** HaloTag-Cbx7[ATLm] in wild-type mES cells (Fractional studies).

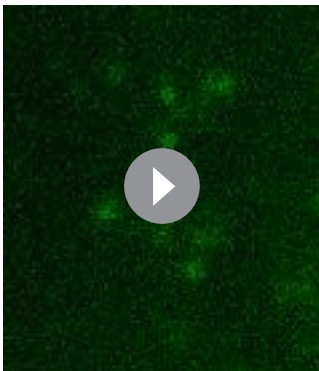

**Video 36.** HaloTag-Cbx7$^{\triangle CD\text{-}ATL}$ in wild-type mES cells (Fractional studies).

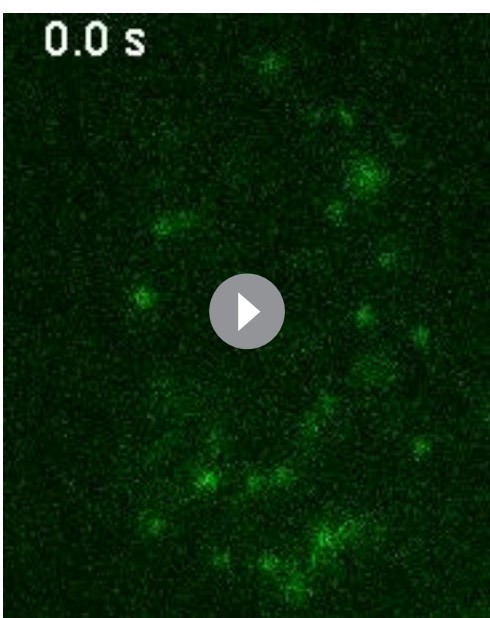

**Video 37.** HaloTag-Cbx7$^{\triangle ATL}$ in wild-type mES cells (Residence time studies).

in mES cells (*Morey et al., 2012*). In contrast to Cbx6, Cbx2 and Cbx4 form the Cbx-PRC1 complex and overlap with H3K27me3 Polycomb domains (*Gao et al., 2012*; *Mardaryev et al., 2016*). Cbx4 is a SUMO E3 ligase and can function as the H3K27me3-dependent or –independent way (*Kagey et al., 2003*; *Li et al., 2014*; *Mardaryev et al., 2016*; *Roscic et al., 2006*).

Our recent study has shown that Cbx2 is targeted to mitotic chromosomes independently of PRC1 and PRC2, and directly recruits the canonical PRC1 components to mitotic chromosomes (*Zhen et al., 2014*). Another study has demonstrated that Cbx2 targets the canonical PRC1 to constitutive heterochromatin by directly recognizing pericentromeric chromatin during early mouse development (*Tardat et al., 2015*). Additionally, *in vitro* study has shown that Cbx2 can directly bind to and compact reconstituted nucleosomes (*Grau et al., 2011*). Thus, these studies suggest additional mechanisms exist to target Cbx2, Cbx4, and Cbx6 to chromatin.

## Targeting of the Cbx7 protein to chromatin by co-recognition of H3K27me3 and DNA

We observed that the level of CD$_{Cbx7}$ at chromatin is less than 30% of Cbx7 and that the residence time of the stable chromatin-bound population of CD$_{Cbx7}$ is about 65% of Cbx7. Mutational analysis demonstrated that Cbx7$^{F11A}$ and Cbx7$^{\triangle CD}$ both remain associating with chromatin. These data imply that additional factor(s) exist(s) to target the Cbx7 protein to chromatin. Cbx7 contains two conserved domains: CD and ATL (*Senthilkumar and Mishra, 2009*). Consistent with previous reports, our data showed that CD$_{Cbx7}$ and ATL$_{Cbx7}$ do not bind to DNA, respectively (*Bernstein et al., 2006*; *Reeves and Nissen, 1990*). Interestingly, our results demonstrated that CD$_{Cbx7}$ and ATL$_{Cbx7}$ together function as a DNA-binding unit. CD-ATL$_{Cbx7}$ exhibited much higher affinity for dsDNA than for ssDNA, dsRNA, and ssRNA. The DNA-binding capacity of CD-ATL$_{Cbx7}$ was functionally significant. Perturbation of the DNA-binding capacity of CD-ATL$_{Cbx7}$ impaired the level of Cbx7 at chromatin and reduced the Cbx7 residence time. Deletion of both CD$_{Cbx7}$ and ATL$_{Cbx7}$ results in the significantly reduced level of the stable chromatin-bound population. Thus, our data demonstrate that the co-recognition of H3K27me3 and DNA by the CD-ATL$_{Cbx7}$ module contributes significantly to the targeting of Cbx7 to chromatin.

Since histone-modifying enzymes typically reside in protein complexes, components within the protein complexes often contribute to targeting of them to chromatin by multivalent engagement of chromatin (*Lalonde et al., 2014*; *Rando, 2012*; *Ruthenburg et al., 2007*). Given that previous studies have shown that Mel18 binds DNA directly *in vitro* (*Akasaka et al., 1996*) and the Ring1b-Mel18 ubiquitin module recognizes reconstituted nucleosome (*McGinty et al., 2014*), we test whether subunits of the Cbx7-PRC1 complex contribute to the binding of Cbx7 to chromatin. Interestingly, our single-molecule approaches demonstrated that depletion of *Ring1a/Ring1b* and *Bmi1/Mel18* results

in the increased chromatin-bound levels and the increased residence times of Cbx7. Further studies are needed to understand whether the Cbx-PRC1 complex formation is required for the targeting specificity of Cbx7.

## Hierarchical cooperation between DNA and H3K27me3

Hierarchical cooperation within chromatin regulatory proteins or complexes between unmodified DNA and histone markers is emerging as a mechanism for gene control. For example, SWR1 is recruited to promoter regions containing nucleosome free region > 50 bp and an adjoining nucleosome by the nanomolar DNA-binding affinity of Swc2, a subunit of SWR1. Once bound, the micromolar affinity of Bdf1 bromodomains for acetylated histones directs SWR1 binding to the +1 nucleosome over the -1 nucleosome (*Ranjan et al., 2013*). Thus, hierarchical cooperation between DNA and histone modifications could underpin the SWR1's role in promoting H2A.Z replacement. Another example is that the Rpd3S histone deacetylase complex binds to H3K36-methylated dinucleosome with 100 pM affinity by multiple engagements of histone modifications and DNA (*Huh et al., 2012*; *Li et al., 2007b*). The DNA- and histone-binding abilities of Eaf3, a subunit of Rpd3S, are self-contained and allosterically regulated by Rco1, another subunit of Rpd3S (*Ruan et al., 2015*).

Our results suggest that the mechanism of targeting of Cbx7 to chromatin is dependent on hierarchical cooperation *via* co-recognition of DNA and H3K27me3 by the CD-ATL$_{Cbx7}$ entity (*Figure 8*). We propose that Cbx7-PRC1 is recruited to chromatin by the CD$_{Cbx7}$ recognition of H3K27me3. We hypothesize that the interaction between H3K27me3 and CD$_{Cbx7}$ triggers conformational changes of the Cbx7-PRC1 complex, which drive the high-affinity interaction between DNA and CD-ATL$_{Cbx7}$. This hypothesis is consistent with our observation that the removal of H3K27me3 significantly reduces the targeting of Cbx7 and Cbx8 to chromatin. Implicit in this model is that the binding of CD-ATL$_{Cbx7}$ to DNA is auto-inhibited by unknown mechanisms and allosterically regulated by the CD$_{Cbx7}$ interaction with H3K27me3. Previous studies have shown that H3K27me3 allosterically activates the methyltransferase activity of the PRC2 complex by its interaction with the C-terminus of Eed (*Jiao and Liu, 2015*; *Margueron et al., 2009*). The allosteric activation facilitates the progression of the H3K27me3 mark on chromatin. Thus, our results unite previous biochemical studies and genetic analysis and provide a novel example of control gene expression *via* integration of genetic DNA and histone modifications. The novel and testable hypothesis should inspire future research of

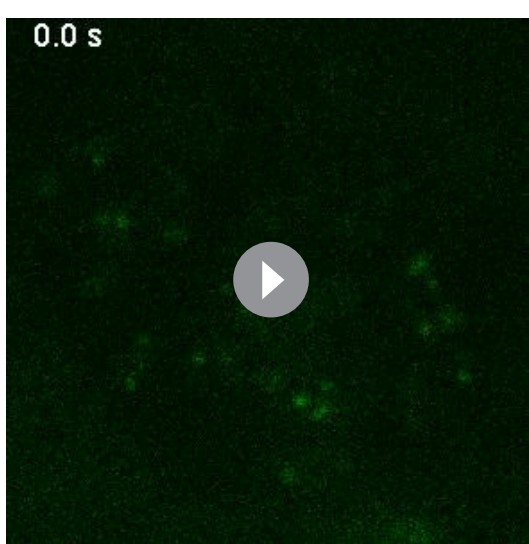

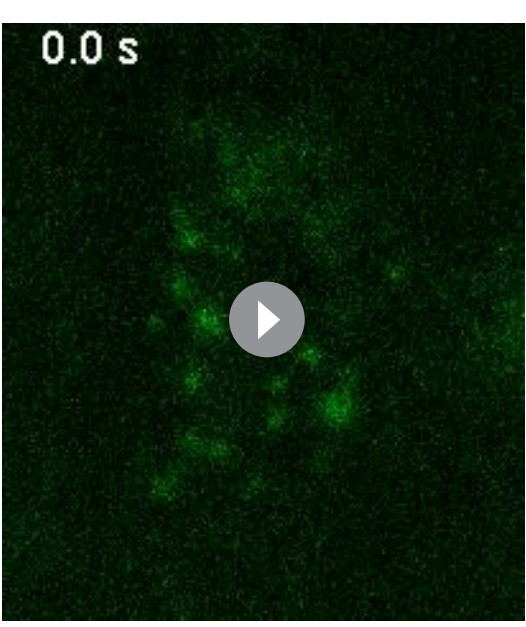

**Video 38.** HaloTag-Cbx7$^{ATLm}$ in wild-type mES cells (Residence time studies).

**Video 39.** HaloTag-Cbx7$^{\triangle CD-ATL}$ in wild-type mES cells (Residence time studies).

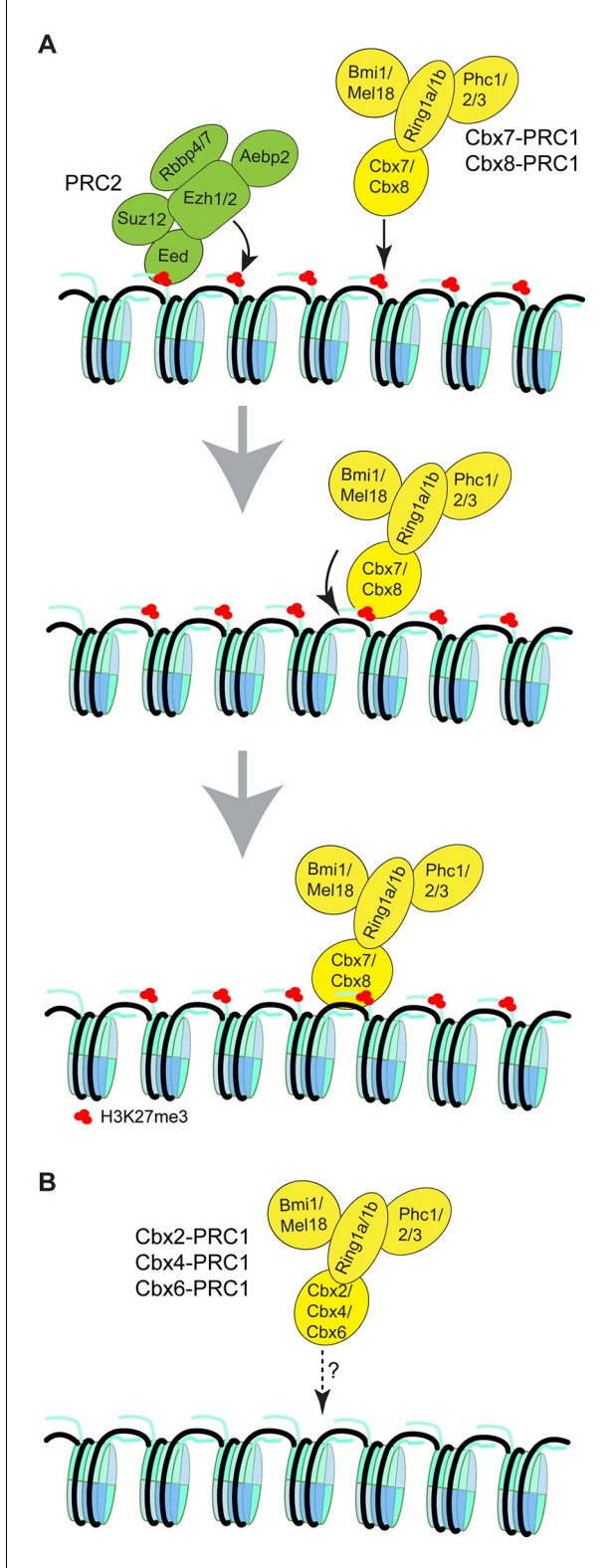

**Figure 8.** Proposed models for the targeting of Cbx-PRC1 to chromatin. (**A**) The Cbx7-PRC1 and Cbx8-PRC1 complexes are targeted to chromatin by co-recognition of H3K27me3 and DNA. The Cbx7- and Cbx8-PRC1 complexes are guided to genomic loci by the CD interaction with H3K27me3. The interaction triggers conformational changes of the Cbx7- and Cbx8-PRC1 complexes and induces the CD-ATL cassette interaction with DNA. Multivalent engagement of DNA and H3K27me3 by the CD-ATL cassette stabilizes the Cbx7- and

*Figure 8 continued on next page*

*Figure 8 continued*
Cbx8-PRC1 complexes at chromatin. (**B**) Molecular mechanisms for the targeting of Cbx2-PRC1, Cbx4-PRC1, and
Cbx6-PRC1 complexes to chromatin remain unknown.

PRC1 function and regulation. The experimental strategy of a combination of genetic engineering, biochemistry, and live-cell SMT should prove widely useful for mechanistic analysis of other chromatin regulatory complexes in living cells.

## Materials and methods

### Cell culture

The $Eed^{-/-}$ (*Endoh et al., 2008*), $Ring1a^{-/-}/Ring1b^{fl/fl}$; $Rosa26::CreERT2$ (*Endoh et al., 2008*), $Bmi1^{-/-}/Mel18^{-/-}$ (*Bmi1* and *Mel18* double knockout) (*Elderkin et al., 2007*) mES cells were obtained from Dr. Haruhiko Koseki (RIKEN Center for Integrative Medical Sciences, Japan). The $Ezh2^{-/-}$ mES cells (*Shen et al., 2009*) were obtained from Dr. Stuart Orkin (Harvard Medical School, Boston, MA). The $Cbx7^{-/-}$ mES cells (*Cheng et al., 2014*) and HEK293T cells were obtained from Dr. Tom Kerppola (University of Michigan Medical School, Ann Arbor, Michigan). The PGK12.1 (*Penny et al., 1996*) mES cells were obtained from Dr. Neil Brockdorff (University of Oxford, UK). The KO mES cell lines have been authenticated by immunoblotting and IF for identity. Other cell lines used in this study have not been authenticated for identity. All cell lines were tested negative for mycoplasma contamination by using DAPI DNA staining. These mES cells were maintained in mES medium (DMEM (D5796; Sigma-Aldrich, St Louis, MO) supplemented with 15% FBS (F0926; Sigma-Aldrich, St Louis, MO), 2 mM glutamine (25030–081; Life Technologies, Carlsbad, CA), 100 units/ml penicillin-streptomycin (15140–122; Life Technologies, Carlsbad, CA), 0.1 mM $\beta$-mercaptoethanol (21985–023; Life Technologies, Carlsbad, CA), $10^3$ units/ml leukemia inhibitor factor (LIF) and 0.1 mM nonessential amino acids (1114050; Life Technologies, Carlsbad, CA)) at 37°C in 5% $CO_2$. 4-hydroxytamoxifen (OHT; H7904; Sigma-Aldrich Inc, St Louis, MO) was administrated to deplete the $Ring1b$ alleles of the $Ring1a^{-/-}/Ring1b^{fl/fl}$; $Rosa26::CreERT2$ mES cells (*Tatavosian et al., 2015*; *Zhen et al., 2014*). HEK293T cells were maintained as described previously (*Tatavosian et al., 2015*; *Zhen et al., 2014*).

### Plasmids

The plasmids pTRIPZ (M)-YFP-Ezh2 and pTRIPZ (M)-YFP-Eed have been described previously (*Tatavosian et al., 2015*). The *YFP* sequence in the plasmids pTRIPZ (M)-YFP-Cbx2 (*Zhen et al., 2014*), pTRIPZ (M)-YFP-Cbx4 (*Zhen et al., 2014*), pTRIPZ (M)-YFP-Cbx6 (*Zhen et al., 2014*), pTRIPZ (M)-YFP-Cbx7 (*Zhen et al., 2014*), pTRIPZ (M)-YFP-Cbx8 (*Zhen et al., 2014*), and pTRIPZ (M)-YFP-H2A (*Zhen et al., 2014*) was replaced with the *HaloTag* sequence amplified from the plasmid ENTR4-HaloTag (w876-1) (Addgene), generating pTRIPZ (M)-HT-Cbx2, pTRIPZ (M)-HT-Cbx4, pTRIPZ (M)-HT-Cbx6, pTRIPZ (M)-HT-Cbx7, pTRIPZ (M)-HT-Cbx8, and pTRIPZ (M)-H2A-HT. The sequence encoding *NLS* (nucleus localization sequence) was chemically synthesized and inserted downstream of the *HaloTag* sequence, generating the pTRIPZ (M)-HT-NLS plasmid. To generate *Cbx7* variants fused with *HaloTag*, the *Cbx7* sequence in the plasmid pTRIPZ (M)-HT-Cbx7 was replaced with the *Cbx7* variant sequences. The Cbx7 variants were as follows: (1) $CD_{Cbx7}$, amino acids 8–62; (2) $Cbx7^{F11A}$, point mutation of F11 to A11; (3) $Cbx7^{\triangle CD}$, deletion of amino acids 1–62; (4) $Cbx7^{ATLm}$, mutation of RKR70-72, K75, and KR77-78 to AGA70-72, A75, and AG77-78); (5) $Cbx7^{\triangle ATL}$, deletion of amino acids 66–83; and (6) $Cbx7^{\triangle CD-ATL}$, deletion of amino acids 1–83. To generate plasmids for expressing *Cbx7* variants in *E.coli*, the sequence encoding the *Cbx7* variants was amplified by PCR and inserted downstream of the *GST* sequence within the pGEX-6P-1-GST vector (GE Healthcare, Pittsburg, PA), generating pGEX-6P-1-GST-$CD_{Cbx7}$ ($CD_{Cbx7}$; amino acids 1–62), pGEX-6P-1-GST-$ATL_{Cbx7}$ ($ATL_{Cbx7}$; amino acids 66–81), pGEX-6P-1-GST-CD-$ATL_{Cbx7}$ (CD-$ATL_{Cbx7}$; amino acids 1–84), and pGEX-6P-1-GST-CD-$ATLm_{Cbx7}$ (CD-$ATLm_{Cbx7}$; amino acids 1–84 with mutation of RKR70-72, K75, and KR77-78 to AGA70-72, A75, and AG77-78). The sequences encoding the fusion genes

have been verified by DNA sequencing and the plasmids will be deposited to Addgene (https://www.addgene.org/).

## Generation of transgenic mES cell lines by lentivirus transduction

Establishing the mES cell lines stably expressing the Polycomb and H2A genes was performed as described previously (*Tatavosian et al., 2015*; *Zhen et al., 2014*). HEK293T cells at 85–90% confluency were co-transfected with 21 µg pTRIPZ (M) containing the fusion gene, 21 µg psPAX2, and 10.5 µg pMD2.G by using calcium phosphate precipitation. At the time of 12 hr after transfection, the medium was replaced with 10 ml DMEM supplemented with 10% FBS, 2 mM L-glutamine, 100 units/ml penicillin G sodium, and 0.1 mM $\beta$-mercaptoethanol. At the time of 48–50 hr after medium change, the medium was harvested to transduce mES cells in the presence of 8 µg/ml polybrene (H9268; Sigma-Aldrich, St Louis, MO) and LIF. For co-transducing multiple genes, lentiviruses were produced separately and mixed at the time of transduction. At the time of 72 hr after transduction, infected cells were selected by using 1.0–2.0 µg/ml of puromycin (P8833; Sigma-Aldrich, St Louis, MO). Unless otherwise indicated, for live-cell single-molecule imaging experiments, the fusions were expressed at the basal level without administrating doxycycline.

## Producing GST-fusion proteins from *E. coli*

Producing GST-Cbx7 fusion proteins is described in more detail at Bio-protocol (*Huynh and Ren, 2017*). The pGEX-6P-1-GST plasmids encoding the *Cbx7* variants were transformed into BL21 competent cells. The gene expression was induced by isopropyl-beta-D-thiogalactopyranoside (AC121; Omega Bio-Tek, Norcross, GA) for 5 hr at 37°C. After centrifugation, the cells were resuspended in cold PBS buffer containing 0.1 mM phenylmethanesulfonyl fluoride (PMSF; 93482; Sigma-Aldrich, St Louis, MO) and protease inhibitor cocktail (P8340; Sigma-Aldrich, St Louis, MO), and then lysed by sonication using Vibra-CellTM sonicator (VCX130; Newtown, CT). To the mixture, 1% Triton X-100 was added. After centrifugation, the supernatant was mixed with pre-washed Glutathione Sepharose beads (17-0756-01; GE Healthcare, Pittsburgh, PA). After 4 times washing with PBS containing 1% Triton X-100, the GST-Cbx7 variant fusion proteins were eluted by 20 mM reduced glutathione (G4251; Sigma-Aldrich, St Louis, MO). After dialysis against PBS, the purity and identity of the GST-Cbx7 variant fusions were assayed by SDS-PAGE.

## EMSA

Alexa Fluor 488-labelled dsDNA-1, dsDNA-2, and ssDNA were purchased from IDT. dsRNA and ssRNA were kindly provided by Dr. Marino Resendiz (University of Colorado Denver). The GST-Cbx7 fusion proteins were mixed with Alexa Fluor 488-labelled dsDNA-1 in binding buffer (20 mM HEPES pH 7.9, 100 mM KCl, 1 mM EDTA pH 8, 5 µM DTT, 0.05 mg/ml bovine serum albumin, and 0.1% NP-40). For competitive assay, DNA and RNA were added to the reaction mixture. After incubation at room temperature for 15 min, 20% glycerol was added to the reaction. The mixtures were then loaded to the wells of Novex 10% Tris-Glycine Mini Protein Gels (EC6075BOX; Life Technologies, Carlsbad, CA). The gels were run for 90 min at 100 V and 400 mA at 4°C in the dark. The gels were imaged using ChemiDoc XRS system (Bio-Rad). The intensities of bands were quantified using ImageJ (http://imagej.nih.gov/ij/).

## IF

IF was performed as described previously (*Tatavosian et al., 2015*; *Zhen et al., 2014*). Wild-type (PGK12.1), $Eed^{-/-}$, $Ezh2^{-/-}$, $Y$-$Eed/Eed^{-/-}$, and $Y$-$Ezh2/Ezh2^{-/-}$ mES cells were cultured on coverslips and fixed using 2.0% paraformaldehyde. After permeabilizing with 0.2% Triton X-100, the cells were washed with basic blocking buffer (10 mM PBS pH 7.2, 0.1% Triton X-100, and 0.05% Tween 20) and then blocked with blocking buffer (the basic blocking buffer plus 3% goat serum and 3% bovine serum albumin). Anti-H3K27me3 antibody (07–449; Millipore, Billerica, MA) was incubated with the cells for 2 hr at room temperature. After washing with the basic blocking buffer, Alexa Fluor 488-labelled goat anti–rabbit antibody (A-11008; Life Technologies, Carlsbad, CA) was incubated with the cells for 1 hr. After incubating with 0.1 µg/ml hoechst, the cells were washed and then mounted on slides with ProLong Antifade reagents (P7481; Life Technologies, Carlsbad, CA). The

images were taken and processed as described previously (*Tatavosian et al., 2015*; *Zhen et al., 2014*).

## Immunoblotting

Immunoblotting was performed as described previously (*Tatavosian et al., 2015*; *Zhen et al., 2014*). Nuclei were lysed in buffer containing 20 mM Tris-HCl pH 7.4, 2.0% NP-40, 500 mM NaCl, 0.25 mM EDTA, 0.1 mM $Na_3VO_4$, 0.1 mM PMSF, and protease inhibitors. Proteins were resolved using NuPAGE 4–12% Bis-Tris Gel (NPO322BOX; Life Technologies, Carlsbad, CA) and transferred to 0.45 μm Immobilon-FL PVDF membrane (IPFL00010; EMD Millipore Corporation, Massachusetts, MA). Membranes were probed with anti-Cbx7 (ab21873; Abcam, MA) and anti-HaloTag (G9281; Promega, Sunnyvale, CA). After incubating with HRP-conjugated anti-rabbit antibody (NA934V; GE Healthcare, Pittsburg, PA), proteins were detected using ECL Plus detection reagents (RPN2106; GE Healthcare, Pittsburg, PA). Membranes were imaged using a ChemiDoc XRS system (BioRad).

## ChIP

ChIP was performed as described previously (*Tatavosian et al., 2015*). *HaloTag-Cbx7/Cbx7$^{-/-}$* and wild-type mES cells were crosslinked with 1.2% formaldehyde (28908; Thermo Fischer Scientific, Waltham, MA) and quenched with 125 mM glycine (G8898; Sigma-Aldrich, St Louis, MO). Nuclei were prepared and washed with LBI buffer (50 mM HEPES pH 7.9, 140 mM NaCl, 1.0 mM EDTA pH 8.0, 10% glycerol, 0.5% NP-40, and 0.25% Triton X-100), LBII buffer (10 mM Tris-HCl pH 8.0, 200 mM NaCl, and 1.5 mM EDTA pH 8.0), and LBIII buffer (10 mM Tris-HCl pH 8.0, 100 mM NaCl, 1.5 mM EDTA pH 8.0, 0.1% sodium deoxycholate, and 0.5% N-lauroylsarcosine). Chromatin was fragmented to 200–1000 bp by sonication (Sonic Vibra Cell model (VCX130)). To the sonicated mixture, 1% Triton X-100 was added. After centrifugation, chromatin was precipitated with anti-HaloTag antibody (G9281; Promega, Sunnyvale, CA). Beads were washed with the RIPA buffer (50 mM HEPES pH 7.9, 500 mM LiCl, 1.0 mM EDTA pH 8.0, 1% NP-40, and 0.7% sodium deoxycholate). The immunoprecipitated DNA were quantified using LightCycler 480 SYBR Green I Master (04707516001; Roche, Indianapolis, IN) with AB Applied Biosystem. Three qPCR replicates were performed. The sequences of the primers used for qPCR have been described previously (*Cheng et al., 2014*; *Ren and Kerppola, 2011*; *Tatavosian et al., 2015*).

## Labelling HaloTag fusion proteins with HaloTag ligand in living cells

Labelling HaloTag Fusion Proteins is described in more detail at Bio-protocol (*Duc and Ren, 2017*). 24 hr prior to imaging, mES cells stably expressing HaloTag fusion proteins were seeded to gelatin-coated cover glass dish. Several concentrations (5 nM, 15 nM, and 30 nM) of Janelia Fluor 549 (JF$_{549}$) HaloTag ligand were used to treat cells for 15 min at 37°C in 5% $CO_2$. Cells were washed with the mES cell medium once and then incubated in the mES cell medium at 37°C in 5% $CO_2$ for 30 min. After replacing with the live-cell imaging medium (A1896701, FluoroBrite DMEM, Life Technologies, Carlsbad, CA), cells were maintained at 37°C using a heater controller (TC-324; Warner Instrument, Hamden, CT) during imaging. Each dish was used for a maximum of 1.5 hr after placing them on the microscope. The number of individual fluorescent spots was typically ~10–50 spots per nucleus by controlling the HaloTag ligand concentration.

## Single-molecule optical setup and image acquisition

Live-cell single molecule tracking was conducted by using a Zeiss Axio Observer D1 Manual Microscopy (Zeiss, Germany) equipped with an Alpha Plan-Apochromatic 100×/1.46 NA Oil-immersion Objective (Zeiss, Germany) and an Evolve 512 × 512 EMCCD camera (Photometrics, Tucson, AZ). Additional magnification of 2.5× was placed on the emission pathway and thus the overall magnification was 250×. The pixel size of the EMCCD was 16 μm. A laser beam from solid state laser (Intelligent Imaging Innovations, CO) was focused on a rotating mirror, which allows to choose wild-field or inclined excitation configuration. The inclined excitation was used to avoid stray-light reflection and reduce background from cell auto-fluorescence (*Tokunaga et al., 2008*). JF$_{549}$ was excited at 552 nm. A Brightline single-band laser filter set (Semrock; excitation filter: FF01-561/14, emission filter: FF01-609/54, and dichroic mirror: Di02-R561-25) was used to filter the excitation and emission wavelength. The microscope and the EMCCD camera were controlled by Slidebook 6.0 software. A

laser power intensity of ~15 mW was used to study diffusion components and a power intensity of ~5 mW for residence times (dissociation constants).

## Single-molecule localization and tracking

U-track algorithm was used for tracking and linking single particles (*Jaqaman et al., 2008*). Before analysis, stacks of images were visually checked and stacks with movement and drift were discarded. About two-thirds of stacks were discarded. The particle localization (x, y) was obtained through 2D Gaussian fitting based on a u-track algorithm using Matlab. A 10-pixel search radius upper limit was allowed for frame-to-frame linking. The detailed localization and tracking parameters were listed in the *Supplementary file 3*. A Matlab script was developed to process the output of 2D tracking from the u-track and to convert the trajectories into a matrix form.

## Extraction of diffusion components

Our SMT was the 2-dimensional projection of the 3-dimensional motion of HaloTag labelled molecules. We assumed that the HaloTag-labelled molecules diffuse isotopically along the three-dimensional axes X, Y, and Z. Thus, the XY projection data reflect the 3-dimensional motion of the molecules. We performed 30-ms integration time without interval. To count labelled molecules from short tracks and to avoid bias toward slowly moving particles that remain visible for longer times, we calculated two kinds of diffusion coefficients: the maximum likelihood diffusion coefficient ($D_m$) per track and the diffusion coefficient of the first step ($D_{f1}$) per track.

$$D_m = \frac{1}{4\tau}\langle r_i^2 \rangle$$

$$D_{f1} = \frac{1}{4\tau} r_{f1}^2$$

where $r_i^2$ and $r_{f1}^2$ are the mean squared step size and the squared first-step size, respectively, and $\tau$ equals 30 ms. The underlying assumption for this analysis was that particles undergo the lateral Brownian motion. An R script was developed to calculate $D_m$ and $D_{f1}$ diffusion coefficients from SMT data (https://gist.github.com/dododas/fb34dc8d9ee5f7d30ebc). The resulting distributions of the logarithm of diffusion coefficients $logD_m$ were pooled from data generated from three independent imaging dishes. We assumed that the chromatin-bound HaloTag-Cbx7 molecules are stationary at chromatin. Thus, the diffusion constant of the chromatin-bound population of the HaloTag-Cbx proteins approximately equals that of the nucleosomal H2A-HaloTag. To estimate the diffusion coefficient of the chromatin-bound component of the HaloTag-Cbx proteins, the distributions of $logD_m$ from the control H2A-HaloTag in wild-type mES cells were fitted with a three-component Gaussian function by OriginLab (OriginLab Corporation).

$$y = y_0 + \sum_{i=1}^{n} \left( \frac{A_i}{w_i \sqrt{\frac{\pi}{2}}} \right) \exp \left( \frac{-2(x - x_i)^2}{w_i^2} \right)$$

where $logD_m$ is offset, $x_i$ is the center of the peak, $A_i$ is the area of the peak, and $w_i$ is the full width at half maximum. The diffusion coefficient of the nucleosomal H2A-HaloTag was determined to be $D_{m1}$ = 0.032 μm²s⁻¹. To systematically compare the CB levels, the subsequent distributions of the HaloTag-Cbx proteins and their variants were fitted with a three-component Gaussian function using the fixed value $D_{m1}$ = 0.032 μm²s⁻¹ while other parameters were set free. There was no convergence if the distributions for HaloTag-NLS in PGK12.1 mES cells and HaloTag-Cbx7 in *Eed*⁻/⁻ mES cells were fitted with a three-component Gaussian function. Thus, a two-component Gaussian function was used for the two distributions. The distributions of the logarithm of diffusion coefficient have previously been used to separate individual populations and to estimate their diffusion coefficients and relative abundance (*Liu et al., 2014*; *Normanno et al., 2015*; *Saxton, 1997*). Fractions of diffusion components were calculated as follows.

$$F_i = \left( \frac{A_i}{\sum_{i=1}^{n} A_i} \right) \times 100\%$$

We denoted the $F_1$ component as the chromatin-bound (CB) population, $F_2$ as the intermediate diffusion (ID) population, and $F_3$ as the fast diffusion (FD) population. Errors were calculated as the s.d. of parameters obtained from fits.

## Determination of residence time

To calculate residence time and survival probability of molecules on chromatin, we performed 30-ms integration time and 170-ms dark time. The track lengths and diffusion coefficients were calculated as described above. We selected molecules for at least two consecutive frames with the maximum likelihood diffusion coefficient $logD_m < 0.10$ µm²/s as chromatin-bound molecules. 97% of H2A-Halo-Tag molecules had diffusion coefficient below this threshold. The duration of individual tracks (apparent residence time) was directly calculated based on the track length. We estimated the residence times of Cbx7 and its variants using the cumulative frequency distribution of dwell times as described in (*Mazza et al., 2012*; *Mazza et al., 2013*; *Morisaki et al., 2014*). To determine the photobleaching rate of JF$_{549}$, mES cells stably expressing H2A-HaloTag were incubated with 500 nM JF$_{549}$ as described above. Live-cell image stacks were taken using the same power and integration and dark time as that for the studying residence times. 9 curves have been obtained. The curves were normalized to 1 and averaged. The averaged curve of photobleaching decay was better described with a two-component exponential decay function based on the F-test implemented in OriginLab.

$$B(\tau) = y_0 + f_{b1}e^{(-\tau/\tau_{b1})} + f_{b2}e^{(-\tau/\tau_{b2})}$$

where $y_0$ is offset, $f_{b1}$ and $f_{b2}$ are amplitude, and $\frac{1}{\tau_{b1}}$ and $\frac{1}{\tau_{b2}}$ are photobleaching rates. The cumulative frequency distributions of dwell times were normalized for photobleaching by dividing by $B(\tau)$ as described in (*Mazza et al., 2012*; *Mazza et al., 2013*; *Morisaki et al., 2014*). The normalized cumulative frequency distributions were better fitted with a two-component exponential decay function based the F-test implemented in OriginLab.

$$y = y_0 + B_1e^{-\tau/\tau_{tb}} + B_2e^{-\tau/\tau_{sb}}$$

where $y_0$ is offset, $B_1$ and $B_2$ are amplitude, and $\tau_{tb}$ and $\tau_{sb}$ are residence times of the transient chromatin-bound component and the stable chromatin-bound component, respectively. Among the chromatin-bound population, fractions of the transient chromatin-bound component ($F_{1tb}$) and the stable chromatin-bound component ($F_{1sb}$) were calculated as follows.

$$F_{1tb} = F_1 \times \frac{B_1}{B_1 + B_2}$$
$$F_{1sb} = F_1 \times \frac{B_2}{B_1 + B_2}$$

where $F_1$ is the chromatin-bound fraction obtained from fitting the distribution of the logarithm of diffusion coefficient.

## Acknowledgements

We thank Dr. Haruhiko Koseki for providing *Eed⁻/⁻*, *Ring1a⁻/⁻/Ring1bfl/fl*; *Rosa26::CreERT2*, and *Bmi1⁻/⁻/Mel18⁻/⁻* mES cell lines, and Dr. Stuart Orkin and Dr. Xiaohua Shen for providing *Ezh2⁻/⁻* mES cell line. We thank Dr. Marino Resendiz for providing RNAs and Dr. Christopher Phiel for sharing instruments. We thank Dr. Aaron M Johnson and Dr. David Engelke for their constructive suggestion and criticism. This work was supported, in whole or in part, by the National Cancer Institute of the National Institutes of Health under Award Number R03CA191443 (to XR). This work was also supported by grants from the CU-Denver Office Research Service (to XR), the American Cancer Society Grant IRG 57-001-53 subaward (to XR), and the Howard Hughes Medical Institute (LDL and JBG).

## Additional information

### Competing interests

JBG, LDL: Filed patent application on the Janelia Fluor (JF) dyes (PCT/US2015/023953). The other authors declare that no competing interests exist.

### Funding

| Funder | Grant reference number | Author |
| --- | --- | --- |
| Howard Hughes Medical Institute | | Jonathan B Grimm<br>Luke D Lavis |
| National Cancer Institute | R03CA191443 | Xiaojun Ren |
| American Cancer Society | IRG 57-001-53 | Xiaojun Ren |
| University of Colorado Denver | Office Research Service | Xiaojun Ren |

The funders had no role in study design, data collection and interpretation, or the decision to submit the work for publication.

### Author contributions

CYZ, Established transgenic mES cell lines, Performed SMT, western blotting and immunofluorescence, Acquisition of data, Analysis and interpretation of data; RT, Performed SMT and ChIP assay, Acquisition of data, Analysis and interpretation of data; TNH, Constructed plasmids, Generated mutants, Established transgenic mES cell lines, Performed EMSA, Acquisition of data, Analysis and interpretation of data; HND, Constructed plasmids, Established mES cell lines, Acquisition of data, Analysis and interpretation of data; RD, Programming support; MK, JL, FJM, Constructed plasmids, Performed SMT, Acquisition of data, Analysis and interpretation of data; JBG, LDL, Contributed reagents, Provided the JF549 dye; YL, Performed SMT, Acquisition of data; TY, Contributed reagents; XR, Conception and design, Acquisition of data, Analysis and interpretation of data, Drafting or revising the article

### Author ORCIDs

Xiaojun Ren, http://orcid.org/0000-0002-3931-7625

## Additional files

### Supplementary files

• Supplementary file 1. Fractional sizes and diffusion constants of the CB, ID, and FD populations obtained from live-cell SMT analysis of the Cbx family proteins and their variants.

• Supplementary file 2. Residence times, transient ($F_{1tb}$) and stable ($F_{1sb}$) chromatin-binding fractions of Cbx7 and its variants.

• Supplementary file 3. U-track parameters used in this research.

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
