## [Decision Letter]

Thank you for submitting your article "Live-Cell Single-Molecule Tracking Reveals Co-recognition of H3K27me3 and DNA Targets Polycomb Cbx7-PRC1 to Chromatin" for consideration by *eLife*. Your article has been reviewed by three peer reviewers, one of whom, Jerry Workman is a member of our Board of Reviewing Editors, and the evaluation has been overseen by Kevin Struhl as the Senior Editor. The following individuals involved in review of your submission have agreed to reveal their identity: Tae-Hee Lee (Reviewer #2).

The reviewers have discussed the reviews with one another and the Reviewing Editor has drafted this decision to help you prepare a revised submission.

Summary:

The Polycomp group proteins are essential for normal embryonic development. Furthermore, the members of Polycomb group are frequently overexpressed or mutated in cancer. Hence, understanding Polycomb action is crucial in health and disease. The Cbx proteins, part of the Polycomb complex, are thought to bind to H3K27me3 modification thereby recruiting the Polycomb protein PRC1 to the chromatin. Although the Cbx-PRC1 recruitment to chromatin is the established model, the molecular mechanisms behind this recruitment is poorly understood. Here, Zhen and colleagues have studied the action of Cbx family proteins by utilizing the new critical single-molecule imaging technique. Their study uncovers a new functional role for Cbx7 in targeting Cbx-PRC1 complex to the chromatin.

The conclusions are qualitatively supported by the data. This is a rare epigenetics paper based on real-time monitoring of nuclear protein behavior. The experiments are in general well-executed. However, significant revisions and clarifications are needed.

Essential revisions:

1) The conclusions need to be restated in a less deterministic manner. Based on my visual inspection of the movie files and the data (diffusion constant histograms), the cases analyzed with two diffusion components still contain a non-negligible chromatin-bound population. I agree with the suggested qualitative trends (i.e. increased or decreased chromatin-bound population). But I disagree with that the data supports the conclusions without uncertainty. I would recommend restating the conclusions with "contributes significantly" or something at a similar level instead of "essential" "necessary" "completely abolished" because these do not reflect the apparent uncertainty in the results.

2) As for data analysis, it is unclear to me why log of the diffusion constant values should distribute normally. According to the classical particle kinetics theory, one diffusion component in these histograms should follow an asymptotic growth (y = 1 – exp(-x/a)). Therefore, the histograms in this manuscript should be fit with three growth functions instead of three normal distributions. Or the authors could explain why log<r^2> should distribute normally. Either way, this point should be addressed before publication.

3) Another point unclear in the analysis is the criteria for a decision on fitting a histogram with two or three diffusion components. The criteria should be clearly stated. If the decisions were not based on quantitative criteria, I recommend fitting all the histograms with three components and drawing the conclusions accordingly (e.g. no convergence with three components or significantly reduced or increased chromatin-bound population, etc.).

4) The authors have shown by depleting H3K27me3 via Eed-/- cells, that chromatin binding of Cbx7 and Cbx8 require H3K27me3 (Figure 2). Cbx7 is further characterized by the authors. The authors should characterize Cbx2, -4, -6 and -8 in Ezh2-/- and Cbx8 in the Eed and Ezh2 rescue cells. This would strengthen the authors conclusions to show that Cbx7, and -8 require H3K27me3, while Cbx2, -4, -6 do not. Especially important would be to indicate that Cbx8 behaves similar to Cbx7.

5) It is not clear why the authors have further characterized Cbx7 but not Cbx8. Previous studies have shown differences in Cbx7 and Cbx8 action (Bernstein et al. 2006, Kaustov et al. 2011). In this manuscript the authors indicate that both Cbx7 and Cbx8 require H3K27me3 for chromatin binding. The authors should clarify why the authors have further characterized Cbx7 but not Cbx8.

6) The authors have measured the dwell times of individual single-molecules and determined the residence time of Cbx7 (Figure 4). Subsequently, the authors show that the residence time of Cbx7 is decreased by mutation or deletion of important Cbx7 regions. The authors state that the dwell time distribution is best fitted to a one-component exponential decay model. However, it is not shown whether the authors have examined the potential fit of their data to a two-, or multi-component exponential decay model.

Recent studies have shown (Chen et al. 2014, Morisaki et al. 2014, Swinstead et al. 2016) that transcription factor dwell times are usually fitted to a two-component exponential decay model, with fast dwell time representing non-specific binding and slow dwell time representing specific binding events at chromatin. The authors should clarify this issue.

See for example, Methods Mol Biol 833:177, 2012

"The survival distribution of bound molecules S(t) allows comparison of different experiments more easily than a traditional histogram of residence times for two reasons: First, in contrast to a traditional histogram, S(t) does not depend on arbitrary choices about the histogram binning. Second, when multiple populations of molecules with different residence times coexist, it is easier to directly visualize the fraction of molecules in each population because the cumulative histogram is fit with a multiexponential decay, with the amplitude of each exponential corresponding to the fraction size."

SMT# 1- For determination of fractional sizes of the three population of molecules, the authors acquire very rapidly (30ms exposure, no interval) and measure the diffusion coefficient of each population with the relative% . The authors discuss two different ways of doing this (Dm and Df1), but the details here are unclear.

SMT #2- For determination of residence times, the authors perform 30ms exposure and 1s interval. Then the authors estimate residence times. It is known that different intervals sample different population of molecules; thus the authors here are measuring different things between the 2 SMT experiments.

One biological contradiction that illustrates this problem is in Figure 7. The mutant Cbx7deltaCD-ATL has no CB (chromatin bound fraction, Figure 7) according to SMT#1, but the authors are able to measure a residence time by SMT#2 for that fraction (Figure 7).

Regarding SMT #2, there several issues in question.

- The exponential fitting of the data seems to be done on the histograms and not with the survival plots. As discussed above, the validity of this method is in question.

- The authors say that the best fit is one-component exponential. However, there is no evidence the authors attempted fitting to two exponentials, and no statistical analysis to justify their conclusion.

- Bleaching correction. Instead of using the same set of data to calculate bleaching, the authors use a fixed sample with histones. This is unacceptable because it is likely that fixed material will behave very differently than the non-fixed. Moreover, their bleaching fitting is again to a 1-exponential, but bleaching decay fits better to 2-exponential decay model. This likely contributes to the unusual residence time numbers the authors are getting. These times look to be a bit fast (~4s) considering the big intervals the authors are using (1s).

7) In the collection of single-molecule data for residence time estimation (Figure 4), the authors have used relative long dark time (Td) of 0.97s with total interval time (Tlap) of 1s. Previous studies have generally used shorter interval times of 10ms (Gebhardt et al. 2013), 20ms^-1^00ms (Mazza et al. 2012), 200ms (Morisaki et al. 2014, Swinstead et al. 2016) and 500ms (Chen et al. 2014). Using a longer interval time will result in the capture of longer over the shorter tracks, which can result into overestimation of residence times. In addition, this could be the reason why the Cbx7 data fits to one-component rather than two-component, as transient binding is not seen due to long interval time. The authors should establish whether long interval time influences their dwell time distribution.

8) In Figure 4, the authors show examples of bound and diffusing Cbx7 molecules. The examples are little puzzling. The bound Cbx7 molecule is stationary in their example for (at least) 70s while their dwell time histogram (Figure 4) ends at 40s. If the authors are capturing 70s track, then it should be seen in the histogram. Furthermore, the diffusing Cbx7 molecule in their example moves within 10s. However, if the residence time of Cbx7 is ~4s, one would expect the molecule to move within 10s. These issues need to be clarified by the authors.

9) The authors show that disruption of Cbx7-PRC1 complex formation facilitates Cbx7 chromatin binding, but does not influence the residence time of Cbx7 (Figure 5). Why does the Cbx7 bind more to chromatin after disruption of complex formation? The authors should discuss this issue in greater depth because as currently presented, there is no explanation on this matter. In addition, in the text authors state that complex formation antagonizes the targeting of Cbx7 to chromatin. However, the data shows that is facilitates the targeting. This should be corrected by the authors.

---

## [Author Response]

*Essential revisions:*

*1) The conclusions need to be restated in a less deterministic manner. Based on my visual inspection of the movie files and the data (diffusion constant histograms), the cases analyzed with two diffusion components still contain a non-negligible chromatin-bound population. I agree with the suggested qualitative trends (i.e. increased or decreased chromatin-bound population). But I disagree with that the data supports the conclusions without uncertainty. I would recommend restating the conclusions with "contributes significantly" or something at a similar level instead of "essential" "necessary" "completely abolished" because these do not reflect the apparent uncertainty in the results.*

We agree with the reviewers (please also see the reply to the point 3). In the revised manuscript, we have restated the conclusions in a less deterministic manner throughout the text (please see the revised manuscript with the track changes).

*2) As for data analysis, it is unclear to me why log of the diffusion constant values should distribute normally. According to the classical particle kinetics theory, one diffusion component in these histograms should follow an asymptotic growth (y = 1 – exp(-x/a)). Therefore, the histograms in this manuscript should be fit with three growth functions instead of three normal distributions. Or the authors could explain why log<r^2> should distribute normally. Either way, this point should be addressed before publication.*

Our analysis assumes several (approximate) normal distributions for log(D) (where D is the diffusion coefficient determined from a single-molecule trajectory). We assume a Brownian particle model where the lateral motion is random such that <r^2> follows a Gaussian distribution, with the relation <r^2> = 4Dτ (Crank, J. (1975) The mathematics of diffusion, *Oxford University Press*; and Dill, K.A., Bromberg, S. (2011) Molecular Driving forces, *Garland Science*). If the longitudinal z-direction is considered, <z> follows an exponential decay (or rise) (Dill, K.A., Bromberg, S. (2011) Molecular Driving forces, *Garland Science*). Our experiment only measures the lateral motion─ the 2-dimensional projection of the 3-dimensional motion of HaloTag labelled molecules. To clarify, in subsection “Extraction of diffusion components”, we have added “The underlying assumption for this analysis was that particles undergo the lateral Brownian motion.”

As for why log(D) distributes normally, roughly speaking this is from the central limit theorem (or Gilbrat's law for the log-normal distribution in particular) applied to a Gaussian process (a sum of independent random variables). One may also use an approximate normal distribution or some other distributions (such as Γ distribution) for D instead of log(D). This does not affect the analysis in the manuscript since the purpose of fitting several log-normal (or other) distributions is to separate individual components, each characterized by its average diffusion constant D and their relative abundance, for mechanistic study.

Early studies have shown that the distributions of logarithm of diffusion coefficient measured from single-particle tracking experiments are Gaussian to be good approximation (Saxton, M.J. (1997) Single-particle tracking: the distribution of diffusion coefficients *Biophysical Journal* 72, 1744). Recently, the fitting of distributions of logarithm of diffusion coefficient with a sum of Gaussian has been employed to separate individual populations and to estimate their D values and relative abundance (Liu, Z. et al. (2014) 3D imaging of *Sox2* enhancer clusters in embryonic stem cells. *ELife* 3; and Normanno, D. et al. (2015) Probing the target search of DNA-binding proteins in mammalian cells using TetR as model searcher *Nat Commun* 6, 7357). In the process of writing the reply letter, the Cech laboratory has also reported to estimate individual populations by using the distribution of logarithm of diffusion coefficient (Cech, T.R. et al. (2016) Live Cell Imaging Reveals the Dynamics of Telomerase Recruitment to Telomeres *Cell*, online). In subsection “Extraction of diffusion components”, we have added “The distributions of logarithm of diffusion coefficient have previously been used to separate individual populations and to estimate their diffusion coefficients and relative abundance (Liu et al., 2014; Normanno et al., 2015; Saxton, 1997).”

*3) Another point unclear in the analysis is the criteria for a decision on fitting a histogram with two or three diffusion components. The criteria should be clearly stated. If the decisions were not based on quantitative criteria, I recommend fitting all the histograms with three components and drawing the conclusions accordingly (e.g. no convergence with three components or significantly reduced or increased chromatin-bound population, etc.).*

We thank the reviewers for this suggestion. We have clarified the criteria in subsection “Extraction of diffusion components”, starting from “We assumed that the chromatin-bound HaloTag-Cbx7 molecules …” Briefly, we assumed that the chromatin-bound HaloTag-Cbx7 molecules are stationary at chromatin. Thus, the diffusion constant of the chromatin-bound population of the HaloTag-Cbx proteins approximately equals that of the nucleosomal H2A-HaloTag. We estimated the diffusion coefficient of the chromatin-bound population of the HaloTag-Cbx proteins from the control H2A-HaloTag in wild-type mES cells. The subsequent distributions of the HaloTag-Cbx proteins and their variants were fitted with a three-component Gaussian function using the fixed value D_m1_ = 0.032 µm^2^s^-1^ while other parameters were set free. Except for HaloTag-NLS in PGK12.1 mES cells and HaloTag-Cbx7 in *Eed^─/─^* mES cells where fits were no convergence if a three-component Gaussian function with the fixed D_m1_ was used, all other distributions were fitted with three components with the fixed D_m1_.

*4) The authors have shown by depleting H3K27me3 via Eed-/- cells, that chromatin binding of Cbx7 and Cbx8 require H3K27me3 (Figure 2). Cbx7 is further characterized by the authors. The authors should characterize Cbx2, -4, -6 and -8 in Ezh2-/- and Cbx8 in the Eed and Ezh2 rescue cells. This would strengthen the authors conclusions to show that Cbx7, and -8 require H3K27me3, while Cbx2, -4, -6 do not. Especially important would be to indicate that Cbx8 behaves similar to Cbx7.*

We have performed SMT of HaloTag-Cbx2, -Cbx4, -Cbx6, and -Cbx8 in *Ezh2^─/─^*mES cells as well as HaloTag-Cbx8 in *YFP-Eed/Eed^─/─^* and *YFP-Ezh2/Ezh2^─/─^* mES cells. These results have been added in Figure 2 and Figure 2—figure supplement 2B-2C. Our results showed that Ezh2 is needed for the binding of Cbx7 and Cbx8 to chromatin, but is not required for Cbx2, Cbx4, and Cbx6. The defective chromatin-bound level of HaloTag-Cbx8 in *Eed^─/─^* and *Ezh2^─/─^* mES cells can be rescued by introducing *YFP-Eed* and *YFP-Ezh2* into their corresponding KO mES cells.

*5) It is not clear why the authors have further characterized Cbx7 but not Cbx8. Previous studies have shown differences in Cbx7 and Cbx8 action (Bernstein et al. 2006, Kaustov et al. 2011). In this manuscript the authors indicate that both Cbx7 and Cbx8 require H3K27me3 for chromatin binding. The authors should clarify why the authors have further characterized Cbx7 but not Cbx8.*

Following the suggestion of the reviewers, in subsection “The Cbx7 CD contributes to, but is not efficient for the targeting of Cbx7 to chromatin”, we have clarified this by adding “In the following studies, we focused on the Cbx7 protein since (1) Cbx7 is smaller than Cbx8 (Figure 1 and Figure 3); (2) Cbx7 contains three conserved domains while Cbx8 has four (Figure 1 and Figure 3); (3) Cbx7-PRC1 is the major canonical PRC1 in mES cells (Morey et al., 2013; Morey et al., 2012); and (4) the expression of Cbx8 is nearly undetectable in mES cells (Morey et al., 2013; Morey et al., 2012).”

*6) The authors have measured the dwell times of individual single-molecules and determined the residence time of Cbx7 (Figure 4). Subsequently, the authors show that the residence time of Cbx7 is decreased by mutation or deletion of important Cbx7 regions. The authors state that the dwell time distribution is best fitted to a one-component exponential decay model. However, it is not shown whether the authors have examined the potential fit of their data to a two-, or multi-component exponential decay model.*

In the revised manuscript, we have studied the residence times by using a 200-ms interval (please also see the reply to the point 7). The goodness of fitting distributions of dwell times is based on the significance of the F-test implemented in OriginLab (OriginLab Corporation). To clarify, in subsection “Determination of residence time”, we have added “The normalized cumulative frequency distributions were better fitted with a two-component exponential decay function based the F-test implemented in OriginLab.”

*Recent studies have shown (Chen et al. 2014, Morisaki et al. 2014, Swinstead et al. 2016) that transcription factor dwell times are usually fitted to a two-component exponential decay model, with fast dwell time representing non-specific binding and slow dwell time representing specific binding events at chromatin. The authors should clarify this issue.*

We thank the valuable insights offered by the reviewers. A similar concern was also raised in the point 7 (please also see the reply to the point 7). In the revised manuscript, we have estimated the residence times of Cbx7 and its variants using a 200-ms interval which allows capturing the transient chromatin-bound population and the stable chromatin-bound population, rather than using a 1-s interval which excludes most of the transient chromatin-bound molecules. The cumulative (survive) distributions of bound molecules were better described with a two-component exponential decay function based the significance of the F-test.

*See for example, Methods Mol Biol 833:177, 2012*

*"The survival distribution of bound molecules S(t) allows comparison of different experiments more easily than a traditional histogram of residence times for two reasons: First, in contrast to a traditional histogram, S(t) does not depend on arbitrary choices about the histogram binning. Second, when multiple populations of molecules with different residence times coexist, it is easier to directly visualize the fraction of molecules in each population because the cumulative histogram is fit with a multiexponential decay, with the amplitude of each exponential corresponding to the fraction size."*

We agree with the reviewers. In the revised manuscript, we have estimated the residence times for individual populations and their relative abundance by using the cumulative (survival) distributions of bound molecules. To clarify, in the revised manuscript subsection “Determination of residence time”, we have added “We estimated the residence times of Cbx7 and its variants using the cumulative frequency distribution of dwell times as described in (Mazza et al., 2012; Mazza et al., 2013; Morisaki et al., 2014)” and “The cumulative frequency distributions of dwell times were normalized for photobleaching by dividing by B(τ) as described in (Mazza et al., 2012; Mazza et al., 2013; Morisaki et al., 2014).”

*SMT# 1- For determination of fractional sizes of the three population of molecules, the authors acquire very rapidly (30ms exposure, no interval) and measure the diffusion coefficient of each population with the relative% . The authors discuss two different ways of doing this (Dm and Df1), but the details here are unclear.*

To clarify this, in the Results section, the following statement has been added.

“The above D_m_ analysis involves averaging over independent pairs of the squared jump distance of a single trajectory with a 30-ms interval. Such averaging might obscure transitions between chromatin-binding, confined, and Brownian motion for the single trajectory of a particle within the observation time. To investigate whether the averaging affects resolving the kinetic fractions, we calculated D_f1_ based on the squared jump distance between the initial position r0 and the first position r1of a single-trajectory with a 30-ms interval, and constructed the log D_f1_ distribution (Figure 1—figure supplement 1).”

*SMT #2- For determination of residence times, the authors perform 30ms exposure and 1s interval. Then the authors estimate residence times. It is known that different intervals sample different population of molecules; thus the authors here are measuring different things between the 2 SMT experiments.*

We agree with the reviewers. In SMT#1, the 30-ms integration time without interval allows capturing the chromatin-bound molecules, the intermediate diffusing molecules, and the fast diffusion molecules. In SMT#2 of the revised manuscript, we performed time-lapse experiments at an integration time, τ_int_, of 30 ms interspersed with a dark time, τ_d_, of 170 ms. Such imaging conditions allow the blurred images of the fast diffusing molecules blending into the background.

*One biological contradiction that illustrates this problem is in Figure 7. The mutant Cbx7deltaCD-ATL has no CB (chromatin bound fraction, Figure 7) according to SMT#1, but the authors are able to measure a residence time by SMT#2 for that fraction (Figure 7).*

Please also see the reply to the point 3. Since there was a non-negligible chromatin-bound population for Cbx7^△CD-ATL^, the distribution of logarithm of diffusion coefficient of Cbx7^△CD-ATL^ was fitted with a three-component Gaussian function. The CB level was estimated to be (8 ± 3)%.

*Regarding SMT #2, there several issues in question.*

*- The exponential fitting of the data seems to be done on the histograms and not with the survival plots. As discussed above, the validity of this method is in question.*

We thank the reviewers again for this suggestion. In the revised manuscript, we have estimated the residence times from the cumulative (survival) distributions of dwell times.

*- The authors say that the best fit is one-component exponential. However, there is no evidence the authors attempted fitting to two exponentials, and no statistical analysis to justify their conclusion.*

To clarify, in the revised manuscript, subsection “Determination of residence time”, we have added “The averaged curve of photobleaching decay was better described with a two-component exponential decay function based on the F-test implemented in OriginLab.” and “The normalized cumulative frequency distributions were better fitted with a two-component exponential decay function based the F-test implemented in OriginLab.”

*- Bleaching correction. Instead of using the same set of data to calculate bleaching, the authors use a fixed sample with histones. This is unacceptable because it is likely that fixed material will behave very differently than the non-fixed. Moreover, their bleaching fitting is again to a 1-exponential, but bleaching decay fits better to 2-exponential decay model. This likely contributes to the unusual residence time numbers the authors are getting. These times look to be a bit fast (~4s) considering the big intervals the authors are using (1s).*

We understood the concern raised by the reviewers. We have determined the photobleaching rate of JF_549_ using living cells. Among 9 photobleaching curves, 4 curves were better described with a one-component exponential decay function and 5 with a two-component exponential decay function based on the significance of the F-test. In the revised manuscript, we normalized individual curves to 1 and averaged all the curves. The averaged curve was better described by a two-component exponential decay function based on the significance of the F-test. We have clarified these in subsection “Determination of residence time”, starting from “To determine the photobleaching rate of JF_549_ …”

A few methods have been developed for correcting photobleaching decay (Mazza, D. et al. (2012) A benchmark for chromatin binding measurements in live cells NAR 40, e119; Gebhardt, J.M.J. et al. (2013) Single-molecule imaging of transcription factor binding to DNA in live mammalian cells Nature Methods 10, 421; and Chen, J. et al. (2014) Single-Molecule Dynamics of Enhanceosome Assembly in Embryonic Stem Cells Cell 156, 1274). We also test the method reported in the Chen’s paper. We fitted the individual photobleaching decay curves by using a one-component exponential decay function. The photobleaching rate was obtained by averaging the individual rates. The cumulative frequency distributions of dwell times were fitted with a two-component decay function, obtaining the apparent residence times for the stable chromatin-bound population and the transient chromatin-bound functions. After photobleaching correction, the residence times and the fractions were comparable to the values reported in the revised manuscript ([Supplementary-material SD8-data]). Thus, our basic conclusions remain unchanged by using two different ways for correcting photobleaching decay.

*7) In the collection of single-molecule data for residence time estimation (Figure 4), the authors have used relative long dark time (Td) of 0.97s with total interval time (Tlap) of 1s. Previous studies have generally used shorter interval times of 10ms (Gebhardt et al. 2013), 20ms^-1^00ms (Mazza et al. 2012), 200ms (Morisaki et al. 2014, Swinstead et al. 2016) and 500ms (Chen et al. 2014). Using a longer interval time will result in the capture of longer over the shorter tracks, which can result into overestimation of residence times. In addition, this could be the reason why the Cbx7 data fits to one-component rather than two-component, as transient binding is not seen due to long interval time. The authors should establish whether long interval time influences their dwell time distribution.*

We thank the insightful comments offered by the reviewers. We agree that using 1-s lapse time will blend the fluorescence signals of both the transient chromatin-bound molecules and the fast diffusing molecules into background. In the revised manuscript, we have performed time-lapse experiments at an integration time, τ_int_, of 30 ms interspersed with a dark time, τ_d_, of 170 ms.

By using a 200-ms interval time, the cumulative distributions of dwell times were better fitted by a two-component exponential decay function based the significance of the F-test, generating the transient chromatin-bound population with residence time 0.7-1.0 s and the stable chromatin-bound population with residence time 4.5-11 s.

The new analysis elaborates the residence times of the transient chromatin-bound population and the stable chromatin-bound population; however the overall conclusion remains the same.

*8) In Figure 4, the authors show examples of bound and diffusing Cbx7 molecules. The examples are little puzzling. The bound Cbx7 molecule is stationary in their example for (at least) 70s while their dwell time histogram (Figure 4) ends at 40s. If the authors are capturing 70s track, then it should be seen in the histogram. Furthermore, the diffusing Cbx7 molecule in their example moves within 10s. However, if the residence time of Cbx7 is ~4s, one would expect the molecule to move within 10s. These issues need to be clarified by the authors.*

We thank the reviewers for carefully checking the figures. In the revised manuscript, new representative examples of bound and diffusing Cbx7 molecules were shown.

*9) The authors show that disruption of Cbx7-PRC1 complex formation facilitates Cbx7 chromatin binding, but does not influence the residence time of Cbx7 (Figure 5). Why does the Cbx7 bind more to chromatin after disruption of complex formation? The authors should discuss this issue in greater depth because as currently presented, there is no explanation on this matter. In addition, in the text authors state that complex formation antagonizes the targeting of Cbx7 to chromatin. However, the data shows that is facilitates the targeting. This should be corrected by the authors.*

We thank the insights offered by the reviewers. By using a 200-ms interval time for estimating residence time, we were able to show that the residence times of Cbx7 in *Ring1a^─/─^/Ring1b^─/─^* and *Bmi1^─/─^/Mel18^─/─^* mES cells are longer than that in wild-type mES cells. Thus, the increased chromatin-bound level of Cbx7 by depletion of *Ring1a/Ring1b* and *Bmi1/Mel18* could be due to the longer residence time.

We have changed “antagonizes” to “facilitates”.